# Efficient optimization accelerator framework for multi-state spin Ising problems

Chirag Garg ⬥ ✉ & Sayeef Salahuddin ✉

Ising Machines are emerging hardware architectures that efficiently solve NP-hard combinatorial optimization problems. Generally, combinatorial problems are transformed into quadratic unconstrained binary optimization (QUBO) form, but this transformation often complicates the solution landscape, degrading performance, especially for multi-state problems. To address this challenge, we model spin interactions as generalized boolean logic function to significantly reduce the exploration space. We demonstrate the effectiveness of our approach on graph coloring problem using probabilistic Ising solvers, achieving similar accuracy compared to state-of-the-art heuristics and machine learning algorithms. It also shows significant improvement over state-of-the-art QUBO-based Ising solvers, including probabilistic Ising and simulated bifurcation machines. We also design 1024-neuron all-to-all connected probabilistic Ising accelerator on FPGA with the proposed approach that shows $\sim 10000\times$ performance acceleration compared to GPU-based Tabucol heuristics and reducing physical neurons by $1.5-4\times$ over baseline Ising frameworks. Thus, this work establishes superior efficiency, scalability and solution quality for multi-state optimization problems.

New computing paradigms are getting significant attention due to exponentially growing computing needs[1]. A wide variety of problems falls into the class of non-deterministic polynomial-time hard (NP-hard) and are difficult to solve optimally using conventional computing solutions[2,3]. The solution space grows exponentially with problem size, therefore making brute force searching impractical for large problem instances. However, heuristics[4] and annealing[2] based approaches have been conventionally used to tackle these computationally hard problems in domains such as logistics[5,6], biology[7], integrated circuits design[8], etc. In this context, Ising machines-based accelerators[9–16] are currently being leveraged to efficiently find the solution to hard optimization problems. Various technologies and hardware architectures have been explored to build these Ising accelerators including quantum annealing with superconducting qubits[17], classical annealing in memristor or RRAM[18,19], coherent Ising machines employing optical oscillators[13,20], coupled oscillators[14,16,21], neuromorphic hardware[22], and stochastic circuits (probabilistic bits)[9,10,12,23]. Quantum annealers show promising results[24] but require low temperatures, making them expensive in cost and power. Therefore, classical alternatives have gained attention due to their room-temperature operation and realization using the current semiconductor process flow. This work particularly focuses on stochastic/probabilistic Ising machines to efficiently solve combinatorial optimization problems.

Probabilistic Ising architectures follow Boltzmann machine binary neural network principles[25,26]. These architectures are physically constructed using binary stochastic neurons interacting with each other and aim to minimize the Ising Hamiltonian (Fig. 1a and Supplementary Note 1). The neuron update follows sigmoidal activation (Fig. 1b) such that neurons stochastically move towards the minimum energy states (Fig. 1c). To harness the efficient solution exploration of these hardware architectures, the cost/energy functions of many complex optimization problems are converted into Ising Hamiltonian form. Based on this conversion, these optimization problems can be broadly divided into three categories (Fig. 1d)[27]. The problem class, comprised of binary solution state space with no imposed constraints, has been

Department of Electrical Engineering and Computer Sciences, University of California, Berkeley, CA, USA. ✉e-mail: chirag_garg@berkeley.edu; sayeef@berkeley.edu

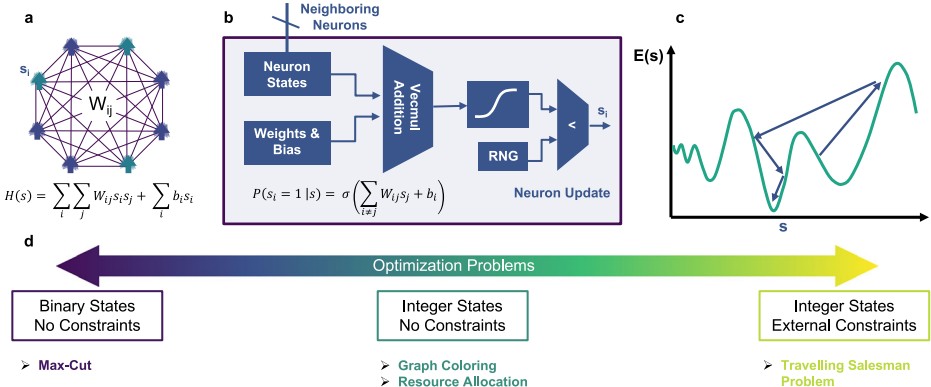

**Fig. 1 | Probabilistic Ising machine architecture and optimization problem classification. a** Quadratic unconstrained binary optimization (QUBO) problems are represented as an Ising network with nodes representing the states and edges depicting interactions between them. **b** Probabilistic Ising architecture based on the Boltzmann machine update rule to minimize the problem Hamiltonian and energy function. **c** The state of the Ising machine evolves and converges to the optimal energy solution. **d** Classification of optimization problems based on the state values and problem constraints.

widely mapped and efficiently solved on Ising machines[16,20,26]. On the other hand, problems with integer/multi-state valued solutions do not naturally convert to binary solution state space, thus leading to additional constraints in the converted Ising Hamiltonian[27–29]. This direct conversion method frequently causes the Ising machines to explore infeasible solution space, resulting in inefficiencies[18,30,31]. One such example is solving the graph coloring problem with Ising machines.

Graph coloring is an NP-hard optimization problem that seeks to assign different colors to the connected nodes of a graph network. It is an example of an integer optimization problem where each integral value represents the color of a node. Previously, annealing and heuristic methods (Tabu-search[32]) were utilized to tackle this problem. These approaches achieve good solution accuracy but suffer from long runtimes for large and densely connected problem instances. Recently, learning-based approaches[33,34] especially graph neural networks are applied to solve the problem accurately and efficiently at a scale. Existing Ising hardware or its modified versions are also proposed to solve this problem. However, they significantly lag behind heuristics and learning-based approaches in solution quality. To an extent, earlier works[18,30,31] overcome this bottleneck by adopting post-processing techniques using additional hardware and software, which adversely affects area and computation time.

In this work, we present an end-to-end probabilistic Ising implementation that combines advances in multi-state problem mapping, spin interaction design, and an efficient hardware architecture to demonstrate significant improvement in area, solution accuracy, and time-to-solution. First, we propose vectorized mapping that represents the node colors as binary vectors rather than using the customary one-hot form. This circumvents the additional mapping constraint in the Ising Hamiltonian that arises due to one-hot encoding[27]. Thus, it completely discards the exploration of infeasible/invalid solution space and improves the solution quality. Second, the interactions among the binary vector states are modeled using truth tables employed in digital logic. Third, we implement an efficient FPGA accelerator that uses higher-order multiplexers to represent these truth tables. Altogether, this multi-state Ising implementation shows approximately 100,000× speed improvement and 5× power improvement compared to its GPU-based implementation for solving problems up to 256 nodes and 16 colors graph coloring problems. This current methodology is then integrated with the parallel tempering approach to improve the solution quality and provide competitive or even better coloring results than Tabucol heuristics and machine learning algorithms.

## Results

### Ising mapping framework

A wide range of combinatorial optimization problems belongs to multistate/integer-valued solution space, including graph coloring, knapsack problems[35], etc. Graph coloring problem, investigated in this work, has applications, such as layout decomposition[36], register allocation[37], logic minimization[38], scheduling[33], and many more. The problem instance that aims to color $N$ nodes with $q$ colors, requires $Nq$ physical neurons/nodes in the Ising framework. Because this framework relies on one-hot encoding to represent node colors, additional constraints are imposed in problem Hamiltonian ($H_A$ in Eq. 1) to enforce the legal combination of one-hot bits[27] (represented by Eq. 2). Therefore, the Ising Hamiltonian ($H$) for graph coloring problem becomes $H_A + H_B$.

$$H_A = A \sum_{i,j \in E} \sum_{k=1}^{q} s_{ik} s_{jk} \tag{1}$$

$$H_B = B \sum_i \left(1 - \sum_{k=1}^{q} s_{ik}\right)^2 \tag{2}$$

In the equations, $s_{ik}$ represents $k^{th}$ one-hot bit for node $i$, $E$ denotes the set of all edges in the problem graph, $A$ is the connectivity weighting factor, and $B$ is the one-hot constraint factor. This soft constraint Hamiltonian is expected to be zero for the ground state energy solution. However, it often takes the non-zero value, thereby making Ising samplers inefficient in solving such problems. (see Fig. 2a). The one-hot encoding requires $q$ bits to represent $q$ possible colors of a node. As a result, Ising machines search over $2^q$ states with only $q$ valid states in the solution space. It makes the optimization hard and deteriorates the solution quality. Hence, the Ising frameworks following one-hot encoding often fail to solve this class of problems. Despite these challenges, this approach is often adopted for most multistate optimization problems because it leads to quadratic interactions supported in most Ising machines[10,16,20,26]. In probabilistic computing[12], binary multiplexers offer simplistic and efficient realization for state-weight multiplication.

### Vectorized mapping framework

In this work, we propose a vectorized mapping approach (Fig. 2b) to tailor the graph coloring problem using a binary encoding technique. It represents the $q$ color state using the $\lceil \log 2q \rceil$ ($= n$) bit vector, which means color for node $S_i$ is represented as $\{s_{i0}, s_{i1}, \ldots, s_{i(n-1)}\}$.

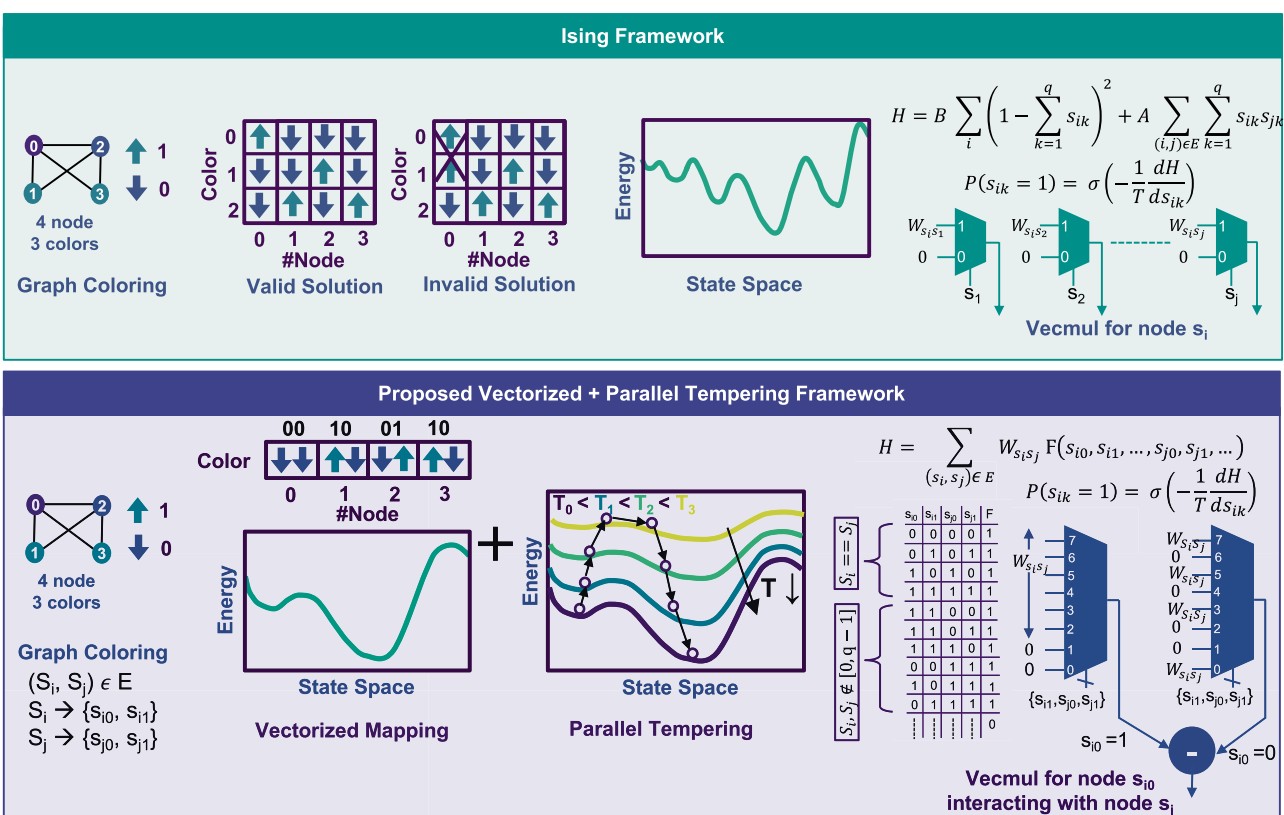

**Fig. 2 | Comparison between Ising and Vectorized mapping approaches.**
**a** Existing Ising framework employs one-hot encoding to map $N$ node $q$ color graph coloring problem to $Nq$ nodes. The encoding constraint is enforced by adding an extra term in the Hamiltonian, creating a complex optimization landscape that can result in suboptimal ground states. This mapping preserves QUBO Hamiltonian, therefore, the interactions are modeled via binary multiplexers in hardware.

**b** Vectorized framework maps the $q$ color state of a node using $\lceil \log 2q \rceil$ hardware nodes. It demonstrates the proposed mapping framework applied for 4 nodes and 3 colors graph coloring problem. This framework searches only valid solutions, helping the hardware reach ground states more easily. Parallel tempering further improves solution exploration. This mapping can be modeled in a truth table format, which is directly mapped to a higher-order multiplexer in hardware.

Therefore, this mapping requires $N\lceil \log 2q \rceil$ physical neurons/nodes for a graph coloring problem with $N$ nodes and $q$ colors. In this way, it eliminates all invalid state space and removes the extra constraints in Ising Hamiltonian (Eq. 2). Therefore, the proposed approach leads to a solvable energy landscape without additional complexities. We also adopt parallel tempering[39] to enhance the solution exploration and improve the solution quality.

$$H = \sum_{(S_i S_j)\in E} W_{S_i S_j} F\left(s_{i0}, s_{i1}, \ldots, s_{i(n-1)}, s_{j0}, s_{j1}, \ldots, s_{j(n-1)}\right) \quad (3)$$

The proposed vectorized mapping is a generic approach that can be applied to any multistate optimization problem being solved on Ising accelerators (see Fig. 2b). This approach represents the states in binary representation and only requires modeling the function operator $F$ in the Hamiltonian $H$ (Eq. 3) for a specific problem. In Eq. 3, $W_{S_i S_j}$ represents the edge ($E$) weight connecting the nodes $S_i$ and $S_j$, and function operator $F$ is modeled in a truth table-based format. Algorithm 1 describes the formulation of the truth table for the graph coloring problem and Fig. 2 shows the formulation of the truth table of 4 nodes, 3-color graph coloring problem. In the cases when nodes ($S_i$ and $S_j$) take the same value or color value greater than $q$ (invalid colors), $F$ becomes one. $F$ is zero for the rest of the cases. The resulting Hamiltonian is a higher-order polynomial due to the $F$ operator function. Ideally, graph coloring criteria are met when the Hamiltonian energy reaches its global minimum of zero, which corresponds to the condition where $F$ is equal to zero. Further, the truth table-based algorithm facilitates its representation as multiplexers, particularly in

the case of Ising hardware, which digitally models spin interactions. In this work, we leverage the probabilistic Ising machines framework to develop the accelerator following the proposed vectorized mapping. The hardware architecture for this scheme is derived from the standard node update rule for probabilistic Ising machines. For $k^{th}$ bit of node $S_i$, the update rule is as follows:

$$P\left(s_{ik} = 1\right) = \sigma\left(-\frac{1}{T}\frac{dH}{ds_{ik}}\right) \quad (4)$$

where $s_{ik}$ takes binary value 0 and 1, $T$ is temperature coefficient and $\sigma$ is sigmoid function ($= 1/(1 + e^{-x})$) for input value $x$. The binary nature of $s_{ik}$ converts the differentiation into $\Delta H$ equal to $H_{s_{ik}=1} - H_{s_{ik}=0}$. This $\Delta H$ is implemented in hardware using higher-order multiplexers followed by a subtraction unit (see Fig. 2), called as vecmul unit. The rest of the hardware implementation remains the same as shown in Fig. 1.

## Accuracy analysis and benchmarks
Next, we investigate the proposed vectorized mapping approach for the graph coloring problem. For benchmarking, we have used it on publicly available COLOR dataset[40]. First, we implement probabilistic Ising frameworks following one-hot encoding (Ising) and proposed binary encoding (vectorized) approaches on GPU for comparison. Figure 3 shows the results for the *queen*13_13 problem, which is deemed a *hard* problem[33]. For both schemes, the exploration for solution happens in a similar way (Fig. 3a). Figure 3b illustrates that the vectorized mapping gives superior coloring results as compared to

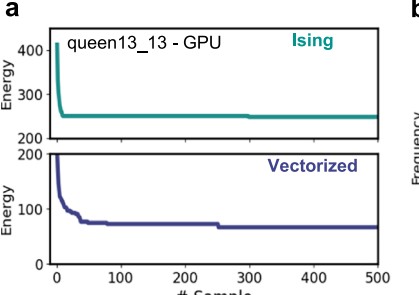
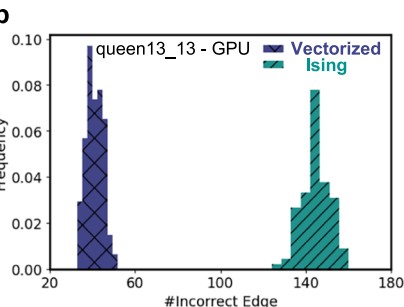
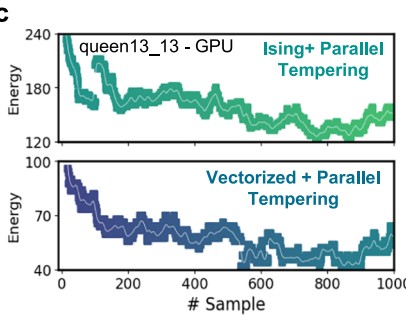

**Fig. 3 | Solution exploration in Ising and Vectorized mapping framework.**
**a** Energy evolution for *queen*13_13 problem instance mapped and solved using Ising and Vectorized mapping implementation on GPU. **b** Distribution of number of incorrectly colored edges achieved for *queen*13_13 problem instance after running GPU-based Ising and Vectorized implementation for 200 parallel runs. **c** Ising and Vectorized framework combined with parallel tempering enhances the solution exploration avoiding the states being stuck around local minima solution.

Ising framework. Even with an optimal set of connectivity weight factor and one-hot constraint weight factor parameters (see Supplementary Fig. S1), the effectiveness of the Ising framework is limited by the one-hot encoding constraint in Eq. 2. This comparison for other problem instances in the dataset is shown in the Supplementary Figs. S3 and S4. Further, we employ the parallel tempering method to enhance the exploration while sampling the energy landscape. This method involves the simultaneous simulation of M replica systems with vectorized mapping running at different temperature schedules. At high temperatures, the system tends to explore the broad region of the solution space, while low temperatures allow precise sampling around a particular region (Fig. 2b). High-temperature replicas ensure that the system doesn't get stuck around the local minimum solution. The states of higher-temperature replica systems are exchanged with low-temperature systems via a swap rule (see Methods Section 1). This state exchange phenomenon enables the exploration of certain parts of solution space that could not be possible via a single temperature schedule. Figure 3c depicts the energy exploration for the *queen*13_13 problem using the parallel tempering approach. It clearly shows that the system continuously escapes from the minimum local energy solution and therefore, achieves the best possible solution accuracy.

We evaluate the accuracy of vectorized mapping solutions against learning-based methods (graph neural network[34] and its GraphSage architecture PI-SAGE[33]), heuristic approaches (Tabucol[32,41]), and state-of-the-art Ising solvers (probabilistic Ising machines[42], simulated bifurcation[43,44]) for graph coloring across standard benchmark datasets. GNN-based solvers frame the coloring problem as a multi-class node classification problem and employ unsupervised training by formulating a loss function. In contrast, the Tabucol technique searches the ground state energy solution by moving small steps and maintains a tabu list to avoid cycling around local minima. We run Tabucol heuristics, probabilistic Ising machines, and vectorized mapping framework with single-flip Gibbs sampling on NVIDIA A100 Tensor Core GPU and solve the same problems 200 times with 1000 iteration (update) steps each. Additionally, simulated bifurcation machines algorithm has been run on the same GPU solving the same problems 200 times with 10000 iteration steps each. We report best possible results achieved by the described methods and compares them in Tables 1 and 2. It includes easy, medium, and hard problem instances labeled in work[45]. The Ising approach performs worse by only being able to solve small and easy problem instances accurately. Among GNN-based solvers, the GraphSage architecture (PI-SAGE) offers better solution accuracy, but it suffers from longer training times[33]. By contrast, proposed vectorized mapping gives competitive coloring results compared to Tabucol heuristics and PI-SAGE GNN while having slightly lower accuracy for hard-to-solve problem instances. Employing the described parallel tempering with vectorized mapping reduces the error up to 50% on these *hard* problems and therefore performs better than other methods.

## Evaluation of success probability and time-to-solution

We employ the success probability metric extensively used in other works[16,46] that captures the solution quality of statistical algorithms. We define error as number of incorrectly colored edges divided by total edges in the problem graph. To calculate success probability ($p_s$), we run each coloring problem for 200 times (or parallel runs) using each algorithm and calculate the success in getting the error less than 2% for those runs. The results in Fig. 4b confirm that the vectorized mapping achieves competitive success probability against the Tabucol heuristic approach and produces a better quality solution compared to state-of-the-art Ising solvers, including probabilistic Ising machine and Simulated Bifurcation machines. Additionally, time-to-solution (TTS) in Eq. 5 is defined as the time needed to obtain a solution within a specified accuracy across multiple runs, with a probability of 99%. $T_{comp}$ represents the average time to complete a single run. The algorithms that achieve success probability greater than 99%, TTS is defined as the average time to reach the solution across parallel runs. Using this methodology, Fig. 4c reports the TTS for different statistical algorithms used to solve the graph coloring problem instances. Overall, the success probability and TTS metric capture a critical aspect of solver performance. A higher success probability reflects a solver's ability to tackle problem instances more efficiently, requiring fewer attempts. This efficiency is represented by incorporating a factor in TTS formulation that accounts for the influence of success probability, providing a robust evaluation of solver effectiveness in terms of both accuracy and computational efficiency.

$$TTS = T_{comp} * \frac{log(1 - 0.99)}{log(1 - p_s)} \tag{5}$$

## FPGA accelerator implementation and results

We implement the proposed vectorized architecture (Fig. 2b) onto VCU118 FPGA to establish the performance acceleration and energy efficiency benchmarks. The FPGA accelerator uses the memory-mapped IO interface used by software applications to program the problem weights and take the solution out (see "Methods" Section 2 for more details). The accelerator (Fig. 1) fetches the weight data from the memory and does neuron states-weight multiplication. This process includes computing the F-operator, defined in Algorithm 1 which is implemented in hardware using truth tables or higher-order multiplexers, as illustrated in Fig. 2b. The structure of this operator is largely determined by the graph's coloring constraints. The accumulated product, represented with 8-bit precision, is passed through a sigmoid

**Table 1 | Solution accuracy in terms of wrongly colored edges (lower value is better) comparison of heuristic (Tabucol), learning-based approaches (GNN and PI-SAGE), Ising methods, and Vectorized framework**

| Problem | #nodes | #edges | #colors | #GNN[33] | #PI-SAGE[33] | #Tabucol | #Probabilistic Ising | #Simulated Bifurcation | #Vectorized GPU | #Vectorized FPGA | # Probabilistic Ising + Parallel Tempering GPU | #Vectorized+ Parallel Tempering GPU |
|---|---|---|---|---|---|---|---|---|---|---|---|---|
| anna | 138 | 493 | 11 | 1 | 0 | 0 | 12 | 44 | 0 | 0 | 2 | 0 |
| david | 87 | 406 | 11 | NA | NA | 0 | 17 | 11 | 0 | 0 | 10 | 0 |
| huck | 74 | 301 | 11 | NA | NA | 0 | 0 | 13 | 0 | 0 | 0 | 0 |
| myciel3 | 11 | 20 | 4 | NA | NA | 0 | 0 | 0 | 0 | 0 | 0 | 0 |
| myciel4 | 23 | 71 | 5 | NA | NA | 0 | 1 | 0 | 0 | 0 | 0 | 0 |
| myciel5 | 47 | 236 | 6 | 0 | 0 | 0 | 0 | 0 | 0 | 0 | 0 | 0 |
| myciel6 | 95 | 755 | 7 | 0 | 0 | 0 | 4 | 6 | 0 | 0 | 2 | 0 |
| myciel7 | 191 | 2360 | 8 | NA | NA | 0 | 144 | 61 | 0 | 0 | 52 | 0 |
| queen5_5 | 25 | 160 | 5 | 0 | 0 | 0 | 5 | 5 | 0 | 0 | 0 | 0 |
| queen6_6 | 36 | 290 | 7 | 4 | 0 | 0 | 3 | 6 | 1 | 1 | 1 | 0 |
| queen7_7 | 49 | 476 | 7 | 15 | 0 | 0 | 21 | 24 | 6 | 5 | 14 | 0 |
| queen8_8 | 64 | 728 | 9 | 7 | 1 | 0 | 41 | 29 | 4 | 2 | 15 | 1 |
| queen9_9 | 81 | 1056 | 10 | 13 | 1 | 0 | 36 | 47 | 5 | 4 | 18 | 2 |
| queen8_12 | 96 | 1368 | 12 | 7 | 0 | 0 | 44 | 73 | 2 | 2 | 21 | 0 |
| queen11_11 | 121 | 1980 | 11 | 33 | 17 | 15 | 87 | 111 | 20 | 18 | 41 | 14 |
| queen13_13 | 169 | 3328 | 13 | 40 | 26 | 21 | 124 | 199 | 31 | 26 | 60 | 21 |

NA is mentioned for problem instances not reported in ref. 33.

**Table 2 | Solution accuracy for citation graphs in terms of wrongly colored edges (lower value is better) comparison of heuristic (Tabucol), learning-based approaches (GNN and PI-SAGE), Ising methods, and Vectorized framework**

| Problem | #nodes | #edges | #colors | #GNN[33] | #PI-SAGE[33] | #Tabucol | #Probabilistic ising | #Simulated bifurcation | #Vectorized GPU | # Probabilistic ising + Parallel Tempering GPU | #Vectorized+ Parallel Tempering GPU |
|---|---|---|---|---|---|---|---|---|---|---|---|
| cora | 2708 | 5278 | 5 | 3 | 0 | 0 | 1004 | 486 | 2 | 830 | 1 |
| citseer | 3279 | 4552 | 6 | 3 | 0 | 0 | 707 | 163 | 1 | 592 | 0 |
| pubmed | 19717 | 44324 | 8 | 35 | 17 | 17 | NA | NA | 18 | NA | 11 |

NA is mentioned for a problem instance that takes more than 24 h to run or when the GPU runs out of resources.

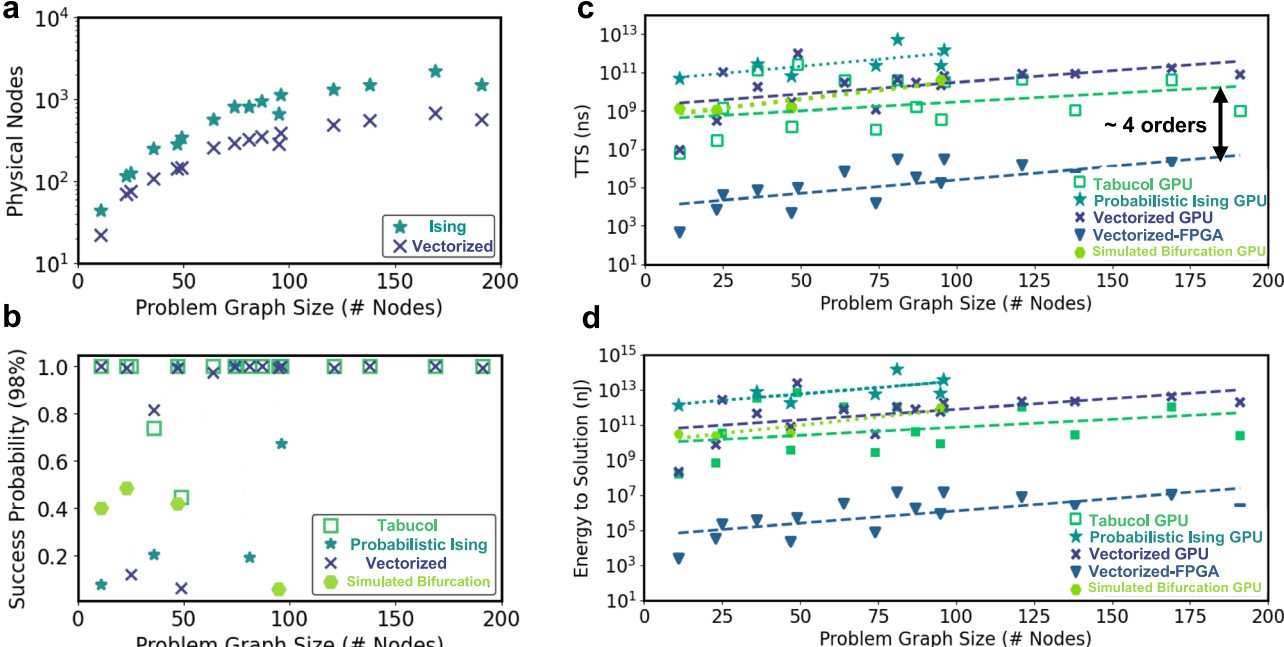

**Fig. 4 | Area, solution quality, performance and energy efficiency benchmarks.** **a** Physical implementation nodes used in Ising and Vectorized mapping approach for dataset problem[40]. **b** Success Probability (>98 % correctly colored edges) metric comparing vectorized mapping with Tabucol heuristics and physics-based Ising solvers including Probabilistic Ising and Simulated Bifurcation Machines. **c** Time-to-solution (TTS) and **d** Energy-to-solution (ETS) benchmark of vectorized mapping implemented on FPGA with Tabucol heuristic and Ising solvers implementation on GPU.

activation function that is implemented using a look-up table (LUT) containing $2^8$ (256) entries. The LUT generates a 16-bit output, which is then compared with a 16-bit random number generated using a Linear Feedback Shift Register (LFSR) to get the updated node value. The architecture runs at 90 MHz clock frequency where only one node gets updated in each cycle. Further, it supports graph coloring problem sizes up to 256 nodes all-to-all connected with a maximum chromatic number of 16, equivalent to 1024 vectorized nodes in total. The framework achieves the same accuracy for dataset problems as the GPU-based vectorized mapping (see Table 1).

The proposed architecture not only gives better accuracy than the baseline Ising-based architecture following one-hot encoding but also reduces the area implementation. The area implementation is quantified in terms of number of physical nodes required to map any problem instance and solve it. The increase in physical nodes leads to an increase in the number of interactions and hence accumulator size grows, leading to a penalty on the implementation area. Figure 4a shows that the vectorized mapping requires 1.5–4 times fewer nodes for the graph coloring problem instances with chromatic numbers up to 16.

The FPGA architecture also uses single flip-Gibbs sampling to update the node, therefore, $N\lceil \log 2(Q)\rceil$ nodes need to be updated for vectorized mapping system for one complete iteration or time step. Thus, the time complexity for the proposed framework will scale with $O(N\lceil \log 2(Q)\rceil)$. The time update for a single node update affects the prefactor of the mentioned time complexity. Specifically, in probabilistic Ising/vectorized, this prefactor is contribution of multiplexer (states-weight multiplication), accumulator, sigmoid activation calculation, and comparator delays. Owing to binary nature of nodes, the accumulator dominates the node update timing[9]. As a result, introducing higher-order multiplexers has minimal impact on the overall time complexity scaling.

The FPGA implementation of the proposed vectorized mapping approach shows ~10,000× speedup compared to Tabucol heuristics and Ising solvers (Probabilistic Ising and Simulated Bifurcation) implementation on NVIDIA A100 Tensor Core GPU (see Fig. 4c).

Vectorized mapping on GPU could not take advantage of efficient multiplication of binary states with weights and, provides only comparable time-to-solution to heuristics. This same trend is expected to hold for any accelerators that are built to support these binary calculations. Similarly, Ising mapping-based solvers, including Probabilistic Ising and Simulated Bifurcation, can benefit from FPGA acceleration[42,47], potentially narrowing the performance gap with our FPGA implementation of the vectorized mapping. Nevertheless, for problems with more than 100 nodes, these solvers may continue to yield sub-optimal solutions. The FPGA accelerator also offers ~5× power improvements over GPU-based vectorized mapping implementation. Therefore, accelerating the time-to-solution by > 4 orders of magnitude on FPGA at a low power budget significantly improves energy efficiency Fig. 4d.

One question that may arise is: is the acceleration achieved for the proposed method over Heuristics solely due to the fact that the vectorized method was implemented on FPGA? In this regard, we note here that the Heuristics take advantage of sequential strategies that do not scale well on parallel architecture. This has been studied extensively in the literature. For example, in Supplementary Fig. S10a we show a comparison between CPU and GPU implementations of Tabucol. It is clearly seen that the GPU implementation provides virtually no acceleration. By contrast GPU implementation of the proposed vectorized method shows large acceleration, as shown in Supplementary Fig. S10b. This underscores a strength of the proposed method that makes it amenable for scaling on specialized hardware such as the FPGA.

## Discussion

Current methods employ the one-hot state encoding framework to map and solve multistate spin Ising problems onto Ising machines. This work presents an alternative approach of vectorized mapping that maps the state using binary encoding. It not only reduces state exploration from $2^{qN}$ to $2^{\lceil \log 2q\rceil N}$ for a $q$ color problem instance but also removes the constraints resulting from the constraint-heavy Ising mapping. As a result, the proposed approach converges close to the

optimal solution achieved by heuristics and learning based approaches. Existing works[12,16,20] on Ising Machines and its derivative frameworks generally aims to achieve accuracy within a certain error range compared to heuristics. However, we show that our method combined with parallel tempering achieves the solution accuracy even better than the heuristics for some of the problems, while retaining the ability to significant speed up on a custom accelerator. We also present a generalized truth table-based method that can be leveraged to map other multi-state problems. Using probabilistic Ising machine architectures, these truth tables to model the neuron interactions can be directly mapped onto higher-order multiplexers. We have implemented this architecture on FPGA to benchmark time-to-solution, energy efficiency, and area efficiency. Overall, the hardware implementation consumes 5 W power and achieves approximately ∼10,000× speedup compared to Tabucol heuristics and ∼100,000× compared to vectorized mapping on GPU. The presented hardware and software framework provides a new way to substantially expand the capabilities of the Ising machines to accurately handle a wide range of multistate optimization problems in a performance and energy-efficient manner.

## Methods
### Parallel tempering
The parallel tempering algorithm utilizes multiple Markov chains (replica chains) running at different temperature schedules representing different probability distributions. It allows a broader exploration of the energy landscape and facilitates better mixing to avoid local energy minima solutions. In this work, we implement 100 parallel chains running at a constant temperature geometrically spaced from temp0 (0.01) and tempn (40). After every 15 sampling steps, adjacent pairs of chains are swapped alternatively with odd-leading and even-leading chain indexes. The update rule is given as:

$$r = exp\left[\left(\frac{1}{T_1} - \frac{1}{T_2}\right)(H(s_{T1}) - H(s_{T2}))\right] \quad (6)$$

$$P_{swap}(s_{T1} - s_{T2}) = \min(1, r) \quad (7)$$

### FPGA setup PCIe
In this work, we implement the vectorized mapping on the Xilinx Virtex UltraScale+ VCU118 FPGA evaluation kit. The FPGA is interfaced with the CPU using Peripheral Component Interconnect Express (PCIe) interface. In particular, we use an open-source Xillybus IP core for the interface with data transfer capabilities of 50MB/s. The data is transferred via a memory-mapped interface implemented using block memory and a designed memory controller.

The digital implementation of 256 nodes and 16 color accelerator supports a network of 1024 probabilistic Ising nodes. These nodes consist of an LFSR-based pseudorandom number generator with a fixed seed, a lookup table-based sigmoidal activation, higher order multiplexer-based matrix multiplication, and a threshold that generates the output state of the neuron. These states are stored in the local memory of FPGA and then transferred to CPU via the PCIe interface.

**Algorithm 1.** Vectorized mapping for graph coloring problem (F-operator):
1: $q$ ← number of colors
2: $H$ ← Hamiltonian Energy
3: $W_{ij}$ ← connectivity weight between node i and j
4: $n$ ← ⌈log 2($q$)⌉
5: $S_i$ ← color vector for node $i\{s_{i0}, s_{i1}, ..., s_{i(n-1)}\}$
6: Generate $F$ operator in truth table format for any two graph nodes ($S_i S_j$):
7: select variables ← $\{s_{i0}, s_{i1}, ..., s_{i(n-1)}, s_{j0}, s_{j1}, ..., s_{j(n-1)}\}$

8: **if** $S_i = S_j$ **then**
9:     $F$ ← 1 [two nodes colored with same color]
10:    $H$ ← $H + W_{ij}$
11: **else if** $S_i, S_j \notin [0, q - 1]$ **then**
12:    $F$ ← 1 [assigned color out of range]
13:    $H$ ← $H + W_{ij}$
14: **else**
15:    $F$ ← 0 [two nodes colored with different color]
16:    $H$ ← $H + 0$
17: **end if**

**Algorithm 2.** Probabilistic Ising Machine Spin Update
1: $H$ ← Hamiltonian
2: $T$ ← temperature
3: $\beta$ ← $1/T$
4: $N_S$ ← Number of iteration (update) steps
5: $k$ ← Total Spins
6: $sigmoid(x) = 1/(1 + e^{-x})$
7: Initialize the spins $s\ \{s_0, s_1, ..., s_k\} \in [0, 1]$ randomly
8: **for** $t = 0$ to $t = N_S$ **do**
9:     **for** $i = 0$ to $i = k$ **do**
10:        Calculate $\Delta H_i = H_{s_i=1} - H_{s_i=0}$
11:        $r$ ← random number
12:        **if** $sigmoid(-\Delta H_i/T) > r$ **then**
13:            $s_i$ ← 1
14:        **else**
15:            $s_i$ ← 0
16:        **end if**
17:    **end for**
18: **end for**

## Data availability
All processed data generated in this study are provided in the main text. The data supporting the plots in this paper can be found in the GitHub repository[48]. Other findings of this study are available from the corresponding authors upon request.

## Code availability
The code used to generate the data in this manuscript can be found in the GitHub[49]

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

## Acknowledgements

This work is supported by the Office of Naval Research (ONR), Multi-disciplinary University Research Initiative (MURI) grant N000142312708.

## Author contributions

C.G. formulated the vectorized scheme, performed the CPU/GPU benchmarks, and designed the vectorized accelerator on FPGA. C.G, and S.S. co-wrote the manuscript; S.S. supervised the research. All authors contributed to discussions and commented on the manuscript.

## Competing interests

The authors declare no competing interests.
