## [Transparent Peer Review file · Nature Communications]

Efficient Optimization Accelerator Framework for Multistate Spin Ising Problems

Corresponding Author: Mr Chirag Garg

Version 0:

Reviewer comments:

Reviewer #1

(Remarks to the Author)

This paper introduces an end-to-end probabilistic Ising implementation featuring multi-state problem mapping and spin interaction design, aimed at reducing the exploration space. While the proposed vector mapping method is interesting, its novelty is quite limited. Below are more detailed comments:

1. The authors claim that "this accelerator solution demonstrates improvement across all metrics over current methods, i.e., energy, performance, area, and solution quality." However, a thorough comparison is necessary. The evaluation should include a range of implementation methods and algorithms to substantiate these claims comprehensively.
2. In this work, the authors encode the one-hot representation into binary vectors, introducing additional encoding and decoding efforts. The overhead associated with these steps should be discussed in the context of different problem types to assess their impact.
3. Binary vector states are represented using truth tables, which can result in significant scalability challenges. The article should address the implications for hardware complexity, time-to-solution, and accuracy as the problem size increases.
4. Table I compares the solution accuracies for solving different graph coloring problems, however, such comparison is unfair. The author should compare "Ising + Parallel Tempering-GPU" and "Vectorized + Parallel Tempering-GPU", otherwise readers cannot figure out the solution accuracy improvement is due to "Vectorized" architecture or "Parallel Tempering" method.
5. The exploration of the Ising and vectorized mapping framework is limited to the queen13_13 problem instance. A broader range of problems, accompanied by statistical analysis, is expected to validate the approach in Fig. 3. Additionally, the observed energy fluctuations in the Ising-only method after state stabilization require explanation.
6. Since parallel tempering is a well-established method applied in multiple Ising machines, the authors are encouraged to reduce the emphasis on results stemming from parallel tempering for accuracy comparison and instead focus on the contributions of the vectorized architecture.
7. The comparison of the proposed FPGA implementation with Tabucol heuristics on an Intel Xeon CPU is unfair. A more appropriate comparison would involve implementing both methods on the same hardware platform to ensure consistency.
8. The claim of ~5x power reduction is questionable since the proposed method is implemented on an FPGA, whereas the comparison group uses a CPU. To make this claim credible, power consumption comparisons should be conducted on the same hardware platform or equivalent platforms.
9. To demonstrate the generality of the vector mapping approach, additional benchmarks beyond the graph coloring problem should be tested. This would provide stronger evidence for the method's applicability across diverse problem domains.

(Remarks on code availability)

Reviewer #2

(Remarks to the Author)

In the manuscript 'Efficient optimization Accelerator framework for multistate Ising problems', the authors propose a novel way to tackle NP-hard optimization problems, specifically graph coloring problems. The main issue with graph coloring and, more general, problems with integer states is that the mapping to a quadratic unconstrained binary problem, QUBO, requires the use of auxiliary variables or spins. Most often, this is done using a one-hot encoding which inflates the size of the problem and makes the energy landscape more rugged. The main result of this manuscript is that the authors, instead of

relying on one-hot encoding, represent a state by a binary-encoded vector. In this way, less spins would be required in implementing e.g. a graph coloring problem; for a problem of size N with q colors $N \log(2q)$ spins are required instead of Nq . The mapping is implemented in a probabilistic Ising machine coded on an FPGA. They report an increase in performance, specifically in solution quality and time-to-solution, and obviously a lower number of physical nodes required to implement the problem. The time-to-solution is only compared with heuristic solvers implemented on CPU.

We believe the manuscript's most important result is the novel encoding. While this is of interest mainly for people developing Ising machines, the paper only explores its effects heuristically on a small set of problems, which begs the question how general these results are. Also, the reliance on the use of a truth table, will make the transition of this technique to hardware implemented and analogue Ising machines (such as the coherent Ising machine) not straightforward and ensures that the technique might be limited to systems that are implemented in software. The implementation on an FPGA is, in itself, not an improvement or a major advance over current Ising machine implementation.

The authors make very bold claims that seem not to be supported by the results such as "... surpassing the capabilities of any existing Ising method." In order for such claim to be validated, one would have expected more thorough comparisons to state of the art Ising methods, such as chaotic amplitude control, Sim-CIM, and others ... The authors have only implemented their encoding on a probabilistic Ising machine and have only compared encodings on this system.

The paper itself is not very accessible due to a lack of detailed explanations of the techniques used and imprecise or even lack of symbol definitions. E.g. the working principle of a probabilistic Ising machine requires much more information than just the update rule. It therefore does not contain sufficient information to replicate the results.

All in all, while the paper does contain interesting results, we believe its impact to be limited within the Ising community and not suited for the broad readership of Nature Communications. The paper lacks detail and description, so we believe that when substantially expanded the paper could find its place in a more dedicated journal. Beyond these general comments, we already give some more detailed considerations that can guide the authors in revising their paper:

- 1) A lot of the explanation of the proposed approach, including the necessary formulas, is done in Figure 2. However, we believe introducing the formulas in the text would improve clarity. Especially as at the moment, a number of symbols (such as σ , E , F and f) are not defined.
- 2) We believe there is a typo in the truth table in Figure 2. We think the last column should be labelled F and not H .
- 3) On page 4, the manuscript mentions that the Hamiltonian is minimized when F is zero. We agree that this is the case when the graph can be colored using the given number of colors, however, this is not the case for all graphs. So that statement should be less general.
- 4) It might be misleading to call the probabilistic Ising architecture 'conventional' as there are a lot of other Ising frameworks out there, each tackling the Ising problem in a different way. Furthermore, as the performance of the novel framework is only compared to the probabilistic Ising architecture, the statement in the abstract that its performance surpasses that of any existing Ising method is too strong.
- 5) An Ising framework typically has a lot of parameters that strongly influence its performance. Could the authors discuss these parameters (their values and whether or not they were optimized in some way) to improve reproducibility. In the same context, could the authors define the 'sigmoid activation function' σ more concretely? The temperature that is chosen and so on ...
- 6) Table 1 reports the solution accuracy in terms of wrongly colored edges for several frameworks. However, it is not clear if this is the average over 200 runs or something else.
- 7) The TTS is not defined with regards to success rate. At least figures comparing success rate and TTS on Ising machines with one hot encoding and vectorized encoding would be necessary and should be discussed in the main text.
- 8) In Figure S2, it is not clear which color represents the Ising framework and which the vectorized framework.
- 9) There are some problems that are shown in Figure S1 that are not mentioned in Table 1
- 10) The colour contrast used in Figures is not suited for colourblind readers.

(Remarks on code availability)

Reviewer #3

(Remarks to the Author)

(Remarks on code availability)

Reviewer #4

(Remarks to the Author)

This paper introduces a novel probabilistic Ising-based accelerator framework for solving multi-state NP-hard combinatorial optimization problems, with a particular focus on graph coloring problems. The authors propose a vectorized mapping approach that is advantageous over traditional one-hot encoding in Ising machines, reducing the solution space from $2^{(qN)}$ to $2^{(\lceil \log_2 q \rceil N)}$ thus improving computational efficiency and solution accuracy. The framework integrates parallel tempering and implements the solution on an FPGA accelerator, claiming up to 10,000× performance acceleration over heuristic methods implemented on a CPU. The approach is benchmarked against graph neural networks (GNNs) and heuristics like Tabucol (run on a CPU), showing superior results in some cases. The work will be of significance to the field of Ising machines.

Major comments:

1) I find it strange that the Ising framework explores completely over invalid solution space when solving the graph coloring problem larger than just 5x5 queen5_5. The authors explained that this framework relies on one-hot encoding implemented with additional constraints defined by Eq.1. When the weighting factor for this constraint is too low, the Ising machine might frequently violate the one-hot encoding constraint, leading to exploration in invalid solution spaces or fail to properly enforce color assignments, causing incorrect graph colorings. On the other hand, if the constraint weighting factor is chosen to be too high, the Ising machine over-prioritizes satisfying the one-hot constraint, making it difficult to optimize for actual coloring constraints. It can also get stuck in poor local minima where valid but non-optimal solutions dominate. Hence, the Ising framework's poor performance may be at least partially due to suboptimal constraint parameters. I encourage the authors to provide more details on how the constraint parameters were chosen. Perhaps, the authors could exploit grid search optimization to identify constraint parameter sets yielding higher solution accuracies and include it in the supplementary materials. That would help to prove that the vectorized mapping approach performs significantly better than the Ising machine framework.

2) I believe that 10,000x acceleration comes mostly from the choice of the platform used for the implementation of the vectorized mapping approach (FPGA) and Tabucol heuristics (CPU). However, it is already well-known that GPU and FPGA platforms allow for significant acceleration at solving compute-intensive problems [1]. For example, there are reports on Tabu search for solving maximum clique problems [4], quadratic assignment problem [3] and hardware/software co-design problems [2]. Given that Tabucol heuristics have been successfully implemented on GPUs and FPGAs (Refs. [2-4]), I encourage the authors to provide a justification for why they did not attempt this implementation. If there are technical constraints preventing a GPU-based Tabucol implementation, a discussion on these limitations would be valuable. Otherwise, a comparison of GPU-/FPGA-accelerated Tabucol vs their FPGA-based vectorized approach would be a fairer benchmark.

3) The paper is very light on details about the truth table operator F. I suggest the authors to add more details on the F operator's construction and size, either in the main text or supplementary materials.

4) The paper does not explicitly clarify whether the F operator must be redefined for each specific graph coloring problem or if it remains the same across different problem instances. If the operator is problem-dependent, it would be useful to discuss how its complexity scales and whether this introduces additional computational overhead compared to conventional mappings, which are typically $O(Nq)$.

Minor comments:

1) To avoid potential misinterpretation, the authors may consider rewording 'brute force searching does not work' to 'brute force searching is impractical for large problem instances' to clarify that brute force is infeasible rather than fundamentally impossible for this class of problems.

2) I encourage the authors to specify whether LUT-based operations introduce any heuristic elements and how large the LUT should be for the 1024-spin IM.

3) To avoid potential confusion, I suggest specifying 'multi-state spin' instead of just 'multi-state' in the title and introduction since 'multi-state' can be perceived as a reference to different configurations of logical spins representing different solutions.

4) The phrase "brute-force Ising approach" may cause some confusion, as it could be misinterpreted to mean that the Ising Machine itself operates via brute-force search. The authors could consider soften it with "constraint-heavy conventional Ising mapping" or clarify that this refers to the inefficiency introduced by excessive constraints in the mapping process, rather than the optimization method itself.

References:

[1] Che, Shuai, Jie Li, Jeremy W. Sheaffer, Kevin Skadron, and John Lach. "Accelerating compute-intensive applications

with GPUs and FPGAs." In 2008 Symposium on Application Specific Processors, pp. 101-107. IEEE, 2008.

[2] Hou, Neng, Fazhi He, Yi Zhou, and Yilin Chen. "An efficient GPU-based parallel tabu search algorithm for hardware/software co-design." *Frontiers of Computer Science* 14 (2020): 1-18.

[3] Novoa, Clara, Apan Qasem, and Abhilash Chaparala. "A SIMD tabu search implementation for solving the quadratic assignment problem with GPU acceleration." In *Proceedings of the 2015 XSEDE Conference: Scientific Advancements Enabled by Enhanced Cyberinfrastructure*, pp. 1-8. 2015.

[4] Kanazawa, Kenji. "FPGA Acceleration of Swap-Based Tabu Search for Solving Maximum Clique Problems." *Journal of Information Processing* 30 (2022): 513-524.

(Remarks on code availability)

Reviewer #5

(Remarks to the Author)

(Remarks on code availability)

Version 1:

Reviewer comments:

Reviewer #1

(Remarks to the Author)

The authors have made revisions based on the reviewers' comments; however, several issues still need to be addressed:

1. The authors have added comparisons with Tabu-search heuristics, machine learning approaches, and a physics-based solver. However, these comparisons are not conducted on a fair basis. For instance, the Tabucol algorithm is implemented on a GPU, the machine learning-based results are taken from the literature, and the vectorized method is implemented on an FPGA. To ensure a fair evaluation, all methods should be tested under the same hardware platform or comparable conditions.
2. The area comparison cannot be solely based on the number of physical nodes, as the hardware complexity per node varies significantly across different implementations. The authors should provide synthesis area metrics (e.g., gate count or layout area) for each method to allow a more meaningful comparison.
3. The authors have included results for solving the Traveling Salesman Problem (TSP). However, addressing instances with only 4 to 14 cities is relatively trivial. The authors are encouraged to tackle more challenging TSP instances and provide detailed descriptions of the simulation setup, including the choice of hyperparameters and any optimization strategies used.
4. The core contribution of this work lies in the proposed encoding method for multi-state Ising problems. However, the significance of this contribution appears limited and should be more critically evaluated or better substantiated.

Overall, the authors are expected to provide more comprehensive details on the simulation procedures to ensure reproducibility. In addition, they should be more cautious with strong claims such as "surpassing the capabilities of any existing Ising method," unless they are rigorously supported by fair and extensive comparisons.

(Remarks on code availability)

Reviewer #2

(Remarks to the Author)

We would like to begin by thanking the authors for their thoughtful response to the initial round of review. The manuscript has improved in several respects, and we appreciate the effort that has gone into the revision. Nevertheless, we remain concerned about the strength and accuracy of certain claims made throughout the paper. In several key areas, the assertions appear overstated and may mislead readers about the true performance and novelty of the proposed approach.

In the abstract, the authors state that their method achieves accuracy comparable to state-of-the-art heuristics and machine learning algorithms, with significant improvements over Ising-based methods. While this is partially supported by Tables 1 and 2, the claim fails to acknowledge a critical trade-off. As illustrated in Figures 4c and 4d, the vectorized method exhibits comparable time-to-solution (TTS) and energy-to-solution to approaches such as Simulated Bifurcation and Tabucol. This suggests that the improved accuracy comes at the cost of longer runtimes, which offsets the overall performance gains. We strongly encourage the authors to revise this claim to more accurately reflect the trade-offs involved and to explicitly discuss this point when introducing Figure 4c.

It is also important to clarify that the reported performance gains are only observed in the FPGA implementation. Figures 4c

and 4d indicate that the performance advantage is largely due to the hardware acceleration provided by the FPGA platform, rather than the proposed encoding scheme itself. Consequently, the claim of a “10,000× performance acceleration over heuristics” is misleading in its current form and requires substantial revision.

Similarly, the statement that parallel tempering reduces coloring error by 50% should be contextualized. This effect is not unique to the proposed method—it applies equally to the Ising-based techniques. Presenting it as a specific advantage of the proposed framework is therefore inaccurate and should be reframed accordingly.

As stated in our previous review report, in our view, the paper’s most promising contribution lies in the novel encoding strategy, which may be of real interest specifically to the Ising machine and optimization community. However, the use of FPGAs for acceleration is well established, and Figures 4c and 4d suggest that the proposed method’s performance and energy efficiency are comparable to existing techniques. While the encoding approach holds potential, the overall results do not meet the threshold of novelty or broad impact expected for publication in Nature Communications. A more specialized journal may be more appropriate if the claims are revised to more accurately reflect the data.

We provide the following detailed suggestions aimed at improving clarity, accuracy, and reproducibility:

* Equation (2): The factor B appears to be missing, and both A and B are only defined in Supplementary Note 4. These definitions should be included in the main text.

* Hyperparameter optimization: The search space is currently limited to a 2D scan over A and B, omitting key parameters such as temperature. This may result in suboptimal performance for the Ising machine and an unfair comparison with the proposed method. Since only the ratio B/A matters (due to energy rescaling), a 1D scan would suffice. The computational resources saved could then be redirected to a broader search that includes temperature and other relevant parameters.

* Definition of E: Just below Equation (2), E is incorrectly defined as the edge between nodes i and j. In fact, E denotes the set of all edges in the graph and should be corrected accordingly.

* Simulated Bifurcation: The description on page 5 lacks sufficient detail. Important parameters are not specified, which impedes reproducibility. Given the sensitivity of this method to parameter choices, additional detail on the optimization process is necessary. Furthermore, its results should be included in Table 2 for completeness.

* Notation consistency: In Supplementary Note 1, the inverse temperature $\beta \sim 1/T$ is used without definition, whereas Supplementary Note 2 refers directly to T. Consistent notation throughout the supplementary materials is essential to avoid confusion.

Conclusion:

In conclusion, while the manuscript introduces an intriguing encoding technique that could stimulate further research, we believe that the current version overstates its performance and novelty. Several of the key claims require significant revision to align with the evidence presented, and issues of clarity and reproducibility remain. Even with these necessary improvements, the manuscript does not meet the threshold of novelty or broad impact expected for Nature Communications.

(Remarks on code availability)

Reviewer #3

(Remarks to the Author)

(Remarks on code availability)

Reviewer #4

(Remarks to the Author)

The authors have improved the manuscript significantly, adding additional benchmarks and expanding the number of instances used for existing benchmarks. The reviewers' concerns and comments were properly addressed.

A new section devoted to benchmarking with the citation graph dataset has been added in the Supplementary Materials. The authors have also added additional benchmarks in Table-1 on page 18 to confirm that the solution accuracy improves mainly due to the vectorized approach.

A grid search parameter optimization has been introduced as suggested. The authors have also provided necessary details about the construction of the truth table operator F in the main text and added an Algorithm-1.

I now fully support the publication of authors' manuscript in Nature Communications.

(Remarks on code availability)

Version 2:

Reviewer comments:

Reviewer #1

(Remarks to the Author)

The authors have addressed my questions in the revised manuscript. I have no further comments.

(Remarks on code availability)

Reviewer #2

(Remarks to the Author)

We appreciate the authors' efforts in addressing our previous comments. It has now become clear to us that there is a misunderstanding surrounding the term heuristics. We interpret heuristics broader as compared to the authors and also include probabilistic Ising machines and simulated bifurcation methods under this category.

Therefore, we had an issue with the "10,000× performance acceleration over heuristics" in the abstract, as the comparison is made solely against Tabucol, and not against the other methods such as simulated bifurcation or probabilistic Ising machines. If our interpretation of what constitutes a heuristic differs from that of the authors, it is likely that other readers may also interpret this differently. Therefore, we think it appropriate if the abstract would be rewritten to be more precise and to be clear that it is a 10,000 acceleration when comparing Tabucol on GPU and the FPGA implementation of the authors.

Additionally, we agree with the authors that it is true that not necessarily all methods will benefit from implementation on the FPGA. Nevertheless, we would like to refer the authors to [R1], which claims that simulated bifurcation can also benefit from an implementation on an FPGA rather than a GPU. Of course, the numbers in this paper cannot necessarily be directly compared with the current manuscript. We do believe that other readers will have the same reaction as us that Figure 4 warrants an apples to apples comparison of all the different methods. Therefore, we think it worthwhile that at least a discussion is included that the performance gap between the proposed vectorized approach on FPGA and e.g. simulated bifurcation on FPGA can be narrower than the factor 10,000.

[R1] K. Tatsumura, A. R. Dixon and H. Goto, "FPGA-Based Simulated Bifurcation Machine," 2019 29th International Conference on Field Programmable Logic and Applications (FPL), Barcelona, Spain, 2019, pp. 59-66, doi: 10.1109/FPL.2019.00019.

(Remarks on code availability)

Reviewer #3

(Remarks to the Author)

(Remarks on code availability)

Re: NCOMMS-24-85810
Efficient Optimization Accelerator Framework for Multistate Ising Problems
Chirag Garg, Sayeef Salahuddin

Reviewer – 1

Summary: This paper introduces an end-to-end probabilistic Ising implementation featuring multi-state problem mapping and spin interaction design, aimed at reducing the exploration space. While the proposed vector mapping method is interesting, its novelty is quite limited.

Author Response

We are thankful to the reviewer for the detailed review of the paper. The comments are valuable to us, and we have modified our manuscript accordingly. Below, we provide a detailed response to reviewer's comment.

Comment 1: The authors claim that "this accelerator solution demonstrates improvement across all metrics over current methods, i.e., energy, performance, area, and solution quality." However, a thorough comparison is necessary. The evaluation should include a range of implementation methods and algorithms to substantiate these claims comprehensively.

Author Response

We appreciate the reviewer's suggestion to conduct a thorough comparison to substantiate the claims made about our accelerator solution. In this work, we use three algorithmic paradigms to benchmark the graph coloring which includes heuristics, machine learning and physics-based solver.

Tabu-search heuristics, such as Tabucol, have been extensively employed to address the graph coloring problem, consistently yielding high-quality solutions [R1-R3]. For fair comparison of solution quality and performance, we changed our CPU implementation of this algorithm to GPU implementation.

Currently, machine learning based approaches are also being employed to tackle such problems to achieve competitive solution accuracy but suffers from long training times [R2]. We benchmark on these algorithms based on publicly available data reported in literature [R2-R3].

On contrary, we propose to use physics-based solvers in particular probabilistic Ising machines to tackle the optimization problems and achieve competitive solution accuracy against previously used approaches at better performance and energy cost. For thorough comparison, we have included additional benchmarks of other Ising methods including state-of-the-art simulated bifurcation machines used to tackle Ising problems [R4-R5].

Solution Quality: To compare solution quality of statistical algorithms which includes heuristics and physics-based Ising solvers, we employ success probability metric extensively used in other works [R6-R7]. We define error as number of incorrectly colored edges divided by total edges in the problem graph. To calculate success probability, we run each coloring problem for 200 iterations (or parallel runs) using each algorithm and calculate the success in getting the error less than 2% for those iterations. The results in Fig. 4b confirms that the vectorized mapping achieve competitive success probability against Tabucol heuristic approach and produces better quality solution compared to state-of-the-art Ising solvers including probabilistic Ising machine and Simulated Bifurcation machines.

We also benchmark our results against learning-based algorithms including graph neural network (GNNs) and its derivative architectures GraphSage (PI-SAGE) which gives deterministic result. Table 1 reports the data comparing absolute coloring error. The proposed vectorized mapping gives competitive coloring results compared to Tabucol heuristics and PI-SAGE GNN while having slightly lower accuracy for hard-to-solve problem instances. However, the integration of parallel tempering with the proposed vectorized mapping yields a notable reduction in error, up to 50%, for these challenging problems. This suggests that the proposed framework, combining vectorized mapping with parallel tempering, offers a promising approach to tackle multi-state spin Ising problems.

Performance: This work extends beyond software version of vectorized mapping by proposing an FPGA-based hardware implementation. Additionally, the FPGA design can be adapted into a custom silicon chip [R8], potentially leading to enhanced energy efficiency and performance. We benchmark the performance of vectorized mapping on FPGA against heuristics (Tabucol) and Ising solvers (Probabilistic Ising and Simulated Bifurcation) implemented on accelerated platform Nvidia A100 Tensor Core GPU. The results in Fig. 4c confirms that the proposed FPGA-based

vectorized implementation shows ~10000X acceleration over heuristics and >10000X over Ising-based solvers.

Energy: Figure 4d illustrates the energy benefits of the vectorized FPGA implementation, which builds upon the improvements shown in Figure 4c. Notably, transitioning from GPU to FPGA hardware yields approximately a five-fold reduction in power consumption, contributing to the overall energy improvements.

Area: The implementation area is quantified in terms of number of physical nodes required to map any problem instance [R5, R8-R9] and solve it on a particular hardware platform. Fig. 4a shows that the vectorized mapping requires 1.5-4 times fewer nodes compared to Ising-based implementation for the graph coloring problem instances.

We believe this expanded evaluation strengthens our findings and provides a more robust validation of our solution's effectiveness.

We have updated the following changes in the main manuscript:

- i) Added additional benchmarks in Fig. 4 on page 17.
- ii) Modified to mentioned line in abstract on Page 1 to be more precise. “Indeed, this accelerator solution achieves superior energy efficiency, performance, area, and solution quality compared to existing methods, including heuristics, machine learning, and Ising machines.”
- iii) Modified the paragraph 2 on page 5 for details on recently added benchmarks.
- iv) Deleted “We implement on Intel Xeon Gold 6330 processor and solve the graph coloring problem instances 200 times with 1000 iteration steps each.” And added Tabucol heuristics with the line 8, paragraph 2, page 5.
- v) Added “Additionally, simulated bifurcation machines algorithm has been run on the same GPU solving the same problems 200 times with 10000 iteration steps each.” line in paragraph 2, Page 5.
- vi) Paragraph 3 on page 5 and paragraph 1 on page 6 have been added to discuss the added success probability metric.
- vii) Added “Ising solvers (Probabilistic Ising and Simulated Bifurcation)” in line 2 paragraph 3 page 7 to capture the result of additional benchmark of simulated bifurcation.

Comment 2: In this work, the authors encode the one-hot representation into binary vectors, introducing additional encoding and decoding efforts. The overhead associated with these steps should be discussed in the context of different problem types to assess their impact.

Author Response

We thank the reviewer for this suggestion. This work utilizes binary vector encoding for optimization problems that comprises of integer states solution. The binary vector representation does not require any additional encoding and decoding efforts, however, only introduces the F-operator in Ising Hamiltonian (Eq. R1). Therefore, the primary task involves formulating the F-operator, as outlined in Algorithm 1.

$$H = \sum_{(S_i, S_j) \in E} W_{S_i S_j} F(S_{i0}, S_{i1}, \dots, S_{j0}, S_{j1}, \dots) \dots\dots\dots(R1)$$

- i) Algorithm-1 has been added on page 20 for clarity.
- ii) A line “This approach represents the states in binary representation and only requires modeling the function operator F in the Hamiltonian (Eq. 2) for a specific problem.” has been added on page 4 paragraph 2 line 2-4.

Comment 3: Binary vector states are represented using truth tables, which can result in significant scalability challenges. The article should address the implications for hardware complexity, time-to-solution, and accuracy as the problem size increases.

Author Response

To demonstrate the scalability of the proposed vectorized approach, we benchmark the citation datasets (Cora [R10], Citeseer [R11], and Pubmed [R12]). These problems are often used for graph-based benchmark experiments and have also been used for testing graph-coloring based algorithms [R2-R3]. Table 2 confirms that the solution accuracy advantage of the proposed vectorized mapping holds even for large size graph problems. Fig. S4 shows the time-to-solution (TTS) for these citation graphs. It confirms that the vectorized mapping on GPU takes around one order of magnitude more time compared to the Tabucol heuristics aligning with the scaling trends in Fig. 4c. Moreover, in probabilistic Ising hardware, the overall hardware scaling is primarily influenced by the number of physical nodes required to solve a problem [R8-R9]. As a result, it follows the scaling behavior of physical nodes with respect to problem size illustrated in Fig. 4a.

A section titled “Vectorized Mapping Benchmark for Citation Graph Dataset” has been added in Supplementary Section 5 on Page 10.

Table-2 on page 19 and Fig. S4 on page 30 have been added for solution accuracy and TTS comparison.

Comment 4: Table I compares the solution accuracies for solving different graph coloring problems, however, such comparison is unfair. The author should compare “Ising + Parallel Tempering-GPU” and “Vectorized + Parallel Tempering-GPU”, otherwise readers cannot figure out the solution accuracy improvement is due to “Vectorized” architecture or “Parallel Tempering” method.

Author Response

We thank the reviewer for pointing this out.

We have added the additional benchmarks in Table-1 on page 18 that includes Probabilistic Ising + Parallel Tempering GPU.

Comment 5: The exploration of the Ising and vectorized mapping framework is limited to the queen13_13 problem instance. A broader range of problems, accompanied by statistical analysis, is expected to validate the approach in Fig. 3. Additionally, the observed energy fluctuations in the Ising-only method after state stabilization require explanation.

Author Response:

Thank you for the suggestion.

The energy exploration and solution analysis of Ising and vectorized mapping framework has been added for other problem instances used for benchmarking in this work has been added in Supplementary Figure S2 and S3 on page 28-29.

Please note that we have modified the Fig. 3a & 3b considering Reviewer#4 suggestion in comment 1. Following up on the suggestion, we exploit grid search parameter optimization for each problem instance separately. This methodological choice enables the Ising solver optimally enforce the one-hot constraint defined in Eq. 1 while determining the problem solutions. Therefore, Fig. 3b compares the number of incorrectly colored edges achieved after completing each of 200 parallel runs instead of valid and invalid energy statistics.

The underlying algorithm used in probabilistic Ising machine which decides the spin update is Gibbs sampling as mentioned in Algorithm R2 [R13-R15]. The update algorithm first calculates the change in Hamiltonian (ΔH) due to spin flip and compares sigmoid($-\Delta H/T$) with a random number from uniform distribution of [0, 1] to accept or reject the spin flip. Even though this comparison predominantly supports the spin flips that cause Hamiltonian to minimize, there is still non-zero probability that it would accept spin flips that cause the Hamiltonian to increase [R14-R17]. It gives rise to the observed energy fluctuation in the energy exploration data.

Algorithm 2 on page 21 has been added for better clarity.

A brief discussion about energy fluctuations has been added in the Supplementary Section 1 page 9 paragraph 2.

Comment 6: Since parallel tempering is a well-established method applied in multiple Ising machines, the authors are encouraged to reduce the emphasis on results stemming from parallel tempering for accuracy comparison and instead focus on the contributions of the vectorized architecture.

Author Response:

We thank the reviewer for this suggestion. We have added additional figure (Fig. 4b) to benchmark solution quality achieved by vectorized mapping without parallel tempering and compare it with heuristics methods and other state-of-the-art Ising methods. However, to provide a comprehensive evaluation, we include results from the combined Vectorized + Parallel Tempering approach only in Table 1 alongside those of other reported methods such as graph neural networks and heuristic algorithms. The combined approach demonstrates competitive, and in some cases, slightly better accuracy compared to other reported methods. This broadens the significance of the work beyond the Ising community, offering valuable insights to researchers using alternative algorithmic paradigms to address similar problems.

Comment 7: The comparison of the proposed FPGA implementation with Tabucol heuristics on an Intel Xeon CPU is unfair. A more appropriate comparison would involve implementing both methods on the same hardware platform to ensure consistency.

Author Response:

We thank the reviewer for pointing this out. Tabu-search heuristics, commonly used in solving complex optimization problems, rely on sequential decision-making processes [R18-R19]. This sequential nature can limit the potential for parallel acceleration, as many heuristics are inherently designed to explore the solution space in a step-by-step manner.

As pointed by the reviewer 4 as well, there have been efforts to parallelize some steps of these algorithms to leverage the power of modern computing architectures like GPUs [R19-R21]. While GPU/FPGA implementation of the heuristic achieves modest speedups of up to an order of magnitude, these improvements are observed primarily for some problem instances. It holds true for our GPU implementation of Tabucol heuristics where we also achieve ~10X improvement for large problem instances and thus improving the scaling curve as shown in Figure R1. It contrasts with the proposed FPGA implementation which achieves almost ~100000X speedup compared to GPU-based vectorized mapping approach. Therefore, comparing GPU-based heuristics with GPU/FPGA implementations is reasonable since the algorithmic nature of heuristics inherently limits their ability to fully exploit modern computing architectures.

Fig. R1 Time-to-solution (ns) comparison for Tabucol heuristics implemented on GPU and CPU.

For all the benchmarks, we updated the results with the GPU-based Tabucol heuristics implementation including Fig. 4 on page 17 and Table 1 on page 18.

Comment 8: The claim of ~5x power reduction is questionable since the proposed method is implemented on an FPGA, whereas the comparison group uses a CPU. To make this claim credible, power consumption comparisons should be conducted on the same hardware platform or equivalent platforms.

Author Response:

We thank the reviewer for pointing this out. The claim of ~5x power reduction is mentioned to compare power consumption of vectorized mapping implementation on FPGA over its GPU implementation.

We have changed the sentence on page 7, paragraph 3, line 7 for better clarity to:

The FPGA accelerator offers ~5x power improvements over GPU-based vectorized mapping implementation.

Comment 9: To demonstrate the generality of the vector mapping approach, additional benchmarks beyond the graph coloring problem should be tested. This would provide stronger evidence for the method's applicability across diverse problem domains.

Author Response

To evaluate the broader applicability of the proposed vectorized mapping framework, we conducted additional experiments on the Traveling Salesman Problem (TSP). Both the probabilistic Ising model and the proposed vectorized mapping were implemented on Nvidia A100 Tensor Core GPU and tested across TSP instances involving up to 14 cities. Each instance was solved 500 times, involving 4000 iteration or update steps per run. As shown in Fig.S7, the proposed vectorized mapping approach achieves higher success probabilities and shorter time-to-solution compared to the Ising model. Additionally, it consistently produces tours that are closer to those found by the well-established Lin–Kernighan (LK) heuristic, indicating improved convergence toward high-quality solutions (see Fig. S8). These results suggest that the vectorized mapping framework is an effective strategy for addressing complex combinatorial optimization problems beyond graph coloring, such as TSP.

A section titled “Traveling Salesman Problem” has been added in Supplementary Section 6 on page 11 and page 12. Fig. S5-S8 has been added on page 31-34 for TSP hyperparameter optimization, success probability, TTS and optimality gap analysis.

Reviewer – 2

Summary: In the manuscript ‘Efficient optimization Accelerator framework for multistate Ising problems’, the authors propose a novel way to tackle NP-hard optimization problems, specifically graph coloring problems. The main issue with graph coloring and, more general, problems with integer states is that the mapping to a quadratic unconstrained binary problem, QUBO, requires the use of auxiliary variables or spins. Most often, this is done using a one-hot encoding which inflates the size of the problem and makes the energy landscape more rugged. The main result of this manuscript is that the authors, instead of relying on one-hot encoding, represent a state by a binary-encoded vector. In this way, less spins would be required in implementing e.g. a graph coloring problem; for a problem of size N with q colors $N \log_2 q$ spins are required instead of Nq . The mapping is implemented in a probabilistic Ising machine coded on an FPGA. They report an increase in performance, specifically in solution quality and time-to-solution, and obviously a lower number of physical nodes required to implement the problem. The time-to-solution is only compared with heuristic solvers implemented on CPU.

Author Response

We thank the reviewer for the detailed highlights of the paper and bringing up a fair point for time-to-solution benchmark. In the light of this comment, we have implemented the Tabucol heuristics on GPU. Additionally, we have included additional Ising methods including probabilistic Ising machine and simulated bifurcation machines [R4-R5] as shown in Fig. 4.

We have updated Fig. 4 based on additional benchmarks on page 17.

Summary: We believe the manuscript’s most important result is the novel encoding. While this is of interest mainly for people developing Ising machines, the paper only explores its effects heuristically on a small set of problems, which begs the question how general these results are. Also, the reliance on the use of a truth table, will make the transition of this technique to hardware implemented and analogue Ising machines (such as the coherent Ising machine) not straightforward and ensures that the technique might be limited to systems that are implemented in software. The implementation on an FPGA is, in itself, not an improvement or a major advance over current Ising machine implementation.

Author Response

We evaluate the proposed encoding method on the coloring dataset [R22], which includes problem instances categorized as easy, medium, and hard [R2]. These instances have been previously used to benchmark various graph coloring algorithms [R1–R3]. The reported coloring errors in Table 1 reflect the varying difficulty of these problems. The consistent improvement in accuracy across all difficulty levels indicates the robustness of the proposed approach.

In response to Reviewer-1's suggestion, we have also included additional benchmarks using large citation graphs in Table 2, where our method continues to demonstrate competitive solution quality. These extended evaluations further support the general applicability of the vectorized mapping approach to tackle graph coloring problem.

As for applicability to Ising machines, there are several implementations that model the spin interactions digitally [R8-R9, R23-R25] where this technique can be used directly. The FPGA implementation demonstrates that this technique is not confined to software and can also be realized on hardware systems. Moreover, this implementation can be translated into an integrated-silicon chip to achieve enhanced benefits in performance and energy similar to [R8].

Summary: The authors make very bold claims that seem not to be supported by the results such as "... surpassing the capabilities of any existing Ising method." In order for such claim to be validated, one would have expected more thorough comparisons to state of the art Ising methods, such as chaotic amplitude control, Sim-CIM, and others ... The authors have only implemented their encoding on a probabilistic Ising machine and have only compared encodings on this system.

Author Response:

Thank you for pointing this out. We have added additional benchmarks of other state-of-the-art Ising method, Simulated Bifurcation Machines [R4] which is shown in Figure 4 and Table-1. Reportedly, it shows similar solution accuracy as probabilistic Ising machine does. This result underscores the core advantage of vectorized mapping over one-hot encoding approach, reducing the solution space from 2^{qN} to $2^{\lceil \log_2 q \rceil N}$.

We have changed the sentence in abstract containing "... surpassing the capabilities of any existing Ising method." for better clarity:

"The proposed methodology achieves similar accuracy compared to state-of-the-art heuristics and machine learning algorithms and demonstrates significant improvement over the state-of-the-art Ising-based approaches, including probabilistic Ising and simulated bifurcation machines."

Summary: The paper itself is not very accessible due to a lack of detailed explanations of the techniques used and imprecise or even lack of symbol definitions. E.g. the working principle of a probabilistic Ising machine requires much more information than just the update rule. It therefore does not contain sufficient information to replicate the results.

Author Response:

We thank the reviewer for the suggestion. We have added additional details describing the probabilistic Ising machines on Page 9 paragraph 2. Algorithm 2 has also been added on page 21 describing the implemented probabilistic Ising framework.

In response to the reviewer's detailed comments below, we have incorporated additional definitions and explanatory details throughout the manuscript.

Summary: All in all, while the paper does contain interesting results, we believe its impact to be limited within the Ising community and not suited for the broad readership of Nature Communications. The paper lacks detail and description, so we believe that when substantially expanded the paper could find its place in a more dedicated journal. Beyond these general comments, we already give some more detailed considerations that can guide the authors in revising their paper:

Author Response:

We appreciate the constructive suggestions to improve the manuscript. As shown in Fig.4 and Table 1, our framework delivers competitive and often superior accuracy at a faster execution time compared to established methods like TabuCol heuristics and GNN-based algorithms. It addresses key limitations by overcoming the scalability issues of existing Ising machines and reducing the complexity of neural network-based approaches. This makes it a practical and efficient solution for real-world applications such as VLSI layout decomposition, resource allocation, logic minimization, and industrial scheduling. In fact, we are not aware of any other implementation (Ising or otherwise) that shows a faster execution compared to heuristic methods at matched accuracy. Thus, this work goes well beyond existing Ising computing implementations and is relevant to a broad community related to optimization problems in general. Therefore, we believe that the work is well suited for Nature Communications.

Comment 1: A lot of the explanation of the proposed approach, including the necessary formulas, is done in Figure 2. However, we believe introducing the formulas in the text would improve clarity. Especially as at the moment, a number of symbols (such as σ , E, F and f) are not defined.

Author Response

Thank you for the suggestion.

- i) We have added Eq. 3 in the main text on page 4.
- ii) Required abbreviations has been added Page 4 paragraph 2 and page 4 paragraph 3.

Comment 2: We believe there is a typo in the truth table in Figure 2. We think the last column should be labelled F and not H.

Author Response

Thank for pointing this out.

We have corrected the type in Fig. 2.

Comment 3: On page 4, the manuscript mentions that the Hamiltonian is minimized when F is zero. We agree that this is the case when the graph can be colored using the given number of colors, however, this is not the case for all graphs. So that statement should be less general.

Author Response

Thank you for bringing this to our attention.

We have corrected the mentioned line by updating it to: “Ideally, graph coloring criteria are met when the Hamiltonian energy reaches its global minimum of zero, which corresponds to the condition where F is equal to zero.” in page 4 paragraph 2 line 10-11.

Comment 4: It might be misleading to call the probabilistic Ising architecture ‘conventional’ as there are a lot of other Ising frameworks out there, each tackling the Ising problem in a different way. Furthermore, as the performance of the novel framework is only compared to the probabilistic Ising architecture, the statement in the abstract that its performance surpasses that of any existing Ising method is too strong.

Author Response

Thank you for the suggestion.

We have removed the use ‘conventional’ word with Ising architectures across the text.

We have added additional benchmarks of other state-of-the-art Ising method, Simulated Bifurcation Machines [R4] which is shown in Figure 4 and Table-1. Reportedly, it shows similar solution accuracy as probabilistic Ising machine does. This result underscores the core advantage of vectorized mapping over one-hot encoding approach, reducing the solution space from 2^{qN} to $2^{\lceil \log 2q \rceil N}$.

Further, we have changed the sentence in abstract containing “... surpassing the capabilities of any existing Ising method.” for better clarity:

“The proposed methodology achieves similar accuracy compared to state-of-the-art heuristics and machine learning algorithms and demonstrates significant improvement over the state-of-the-art Ising-based approaches, including probabilistic Ising and simulated bifurcation machines.”

Comment 5: An Ising framework typically has a lot of parameters that strongly influence its performance. Could the authors discuss these parameters (their values and whether or not they were optimized in some way) to improve reproducibility. In the same context, could the authors define the ‘sigmoid activation function’ σ more concretely? The temperature that is chosen and so on ...

Author Response

Thank you for the suggestion.

We have added supplementary section 4 on page 10 and Fig. S1 on page 27 to discuss the hyperparameter optimization. We have defined Sigmoid activation function on page 4 paragraph 3 line 1-2 and in Algorithm 2 on page 21.

Comment 6: Table 1 reports the solution accuracy in terms of wrongly colored edges for several frameworks. However, it is not clear if this is the average over 200 runs or something else.

Author Response

Thank you for pointing this out. Table 1 reports best possible solution accuracy achieved by the machine learning, heuristics and Ising-based methods. It is done to maintain consistency with existing work and to enable fair benchmarking against them [R2-R3]. However, we have added Fig. S2 showing the statistical distribution of wrongly colored edges and success probability metric results in Fig. 4b to statistically compare the algorithms as well.

For better clarity, we have added the line “We report best possible results achieved by the described methods and compares them in Table 1.” on page 5 paragraph 2 line 12.

Fig. S2 on page 28 and Fig. 4b on page 17 have been added to statistically compare the algorithms.

Comment 7: The TTS is not defined with regards to success rate. At least figures comparing success rate and TTS on Ising machines with one hot encoding and vectorized encoding would be necessary and should be discussed in the main text.

Author Response

We thank the reviewer for bringing this up.

We have added Fig. 4b and Fig. 4c on page 17 comparing success probability and TTS metric for different algorithms used in this work.

Additionally, the following paragraph has been added on page 5 and page 6.

“We employ the success probability metric extensively used in other works [16, 46] that captures the solution quality of statistical algorithms. We define error as number of incorrectly colored edges divided by total edges in the problem graph. To calculate success probability (p_s), we run each coloring problem for 200 times (or parallel runs) using each algorithm and calculate the success in getting the error less than 2% for those iterations. The results in Fig. 4b confirms that the vectorized mapping achieve competitive success probability against Tabucol heuristic approach and produces better quality solution compared to state-of-the-art Ising solvers including probabilistic Ising machine and Simulated Bifurcation machines. Additionally, time-to-solution (TTS) in Eq. 5 is defined as the time needed to obtain a solution within a specified accuracy across multiple runs, with a probability of 99%. T_{comp} represents the average time to complete a single run. The algorithms that achieve success probability greater than 99%, TTS is defined as the average time to reach the solution across parallel runs. Using this methodology, Fig. 4c reports the TTS for different statistical algorithms used to solve the graph coloring problem instances. Overall, the success probability and TTS metric capture a critical aspect of solver performance. A higher success probability reflects a solver’s ability to tackle problem instances more efficiently, requiring fewer attempts. This efficiency is represented by incorporating a factor in TTS formulation that accounts for the influence of success probability, providing a robust evaluation of solver effectiveness in terms of both accuracy and computational efficiency.”

Comment 8: In Figure S2, it is not clear which color represents the Ising framework and which the vectorized framework.

Author Response

Thank you for the suggestion.

Fig. S2 has been moved to Fig. S3 on page 29, and we have added the appropriate labels in the figure.

Comment 9: There are some problems that are shown in Figure S1 that are not mentioned in Table 1

Author Response

Thank you for pointing this out. We have added all the problems in Table 1 on page 18.

Comment 10: The colour contrast used in Figures is not suited for colourblind readers.

Author Response

Thank you for pointing this out.

We have changed out plots by using appropriate symbols and labels for better readability.

Summary: This paper introduces a novel probabilistic Ising-based accelerator framework for solving multi-state NP-hard combinatorial optimization problems, with a particular focus on graph coloring problems. The authors propose a vectorized mapping approach that is advantageous over traditional one-hot encoding in Ising machines, reducing the solution space from $2^{(qN)}$ to $2^{([\log_2 q]N)}$ thus improving computational efficiency and solution accuracy. The framework integrates parallel tempering and implements the solution on an FPGA accelerator, claiming up to 10,000× performance acceleration over heuristic methods implemented on a CPU. The approach is benchmarked against graph neural networks (GNNs) and heuristics like Tabucol (run on a CPU), showing superior results in some cases. The work will be of significance to the field of Ising machines.

Author Response

We thank the reviewer for acknowledging the key aspects of proposed vectorized mapping, i.e., reducing solution space that results in improving computational efficiency and solution accuracy. Furthermore, we greatly appreciate the following detailed comments, and we have modified our manuscript accordingly.

Comment 1: I find it strange that the Ising framework explores completely over invalid solution space when solving the graph coloring problem larger than just 5x5 queen5_5. The authors explained that this framework relies on one-hot encoding implemented with additional constraints defined by Eq.1. When the weighting factor for this constraint is too low, the Ising machine might frequently violate the one-hot encoding constraint, leading to exploration in invalid solution spaces or fail to properly enforce color assignments, causing incorrect graph colorings. On the other hand, if the constraint weighting factor is chosen to be too high, the Ising machine over-prioritizes satisfying the one-hot constraint, making it difficult to optimize for actual coloring constraints. It can also get stuck in poor local minima where valid but non-optimal solutions dominate. Hence, the Ising framework’s poor performance may be at least partially due to suboptimal constraint parameters. I encourage the authors to provide more details on how the constraint parameters were chosen. Perhaps, the authors could exploit grid search optimization to identify constraint parameter sets yielding higher solution accuracies and include it in the supplementary materials. That would help to prove that the vectorized mapping approach performs significantly better than the Ising machine framework.

Author Response

The reviewer hits on a very critical point that the Ising framework requires an optimal balance connectivity weight factor and one-hot constraint weight factor. To address this, we optimize these parameters separately for each problem instances using a grid search, as recommended. For each problem instance, we run the probabilistic Ising framework for 1000 iterations and select the parameter set that yields the lowest coloring error (see Fig. S1). As shown in the updated Table 1 and Fig. S2, our proposed vectorized mapping approach outperforms the Ising machine framework. Therefore, even with an optimal set of parameters, the effectiveness of the Ising framework is limited by the one-hot encoding constraint in Eq. (2).

We have updated the following changes in the main manuscript:

- i) Supplementary section 4 on page 10 is added for hyperparameter optimization.
- ii) Fig. S1 has been added on page 27 showing results of grid-search optimization.
- iii) Instead of reporting valid-invalid energy states, we changed the plots to incorrectly colored edges in Fig. 3 on page 16 and Fig. S2 on page 28.
- iv) The following lines, “Fig. 3b illustrates that the vectorized mapping gives superior coloring results as compared to Ising framework. Even with an optimal set of connectivity weight factor and one-hot constraint weight factor parameters (see Fig.S1), the effectiveness of the Ising framework is limited by the one-hot encoding constraint in Eq.2.” has been added on page 5 paragraph 1 line 2-5.

Comment 2: I believe that 10,000x acceleration comes mostly from the choice of the platform used for the implementation of the vectorized mapping approach (FPGA) and Tabucol heuristics (CPU). However, it is already well-known that GPU and FPGA platforms allow for significant acceleration at solving compute-intensive problems [1]. For example, there are reports on Tabu search for solving maximum clique problems [4], quadratic assignment problem [3] and hardware/software co-design problems [2]. Given that Tabucol heuristics have been successfully

implemented on GPUs and FPGAs (Refs. [2-4]), I encourage the authors to provide a justification for why they did not attempt this implementation. If there are technical constraints preventing a GPU-based TabuCol implementation, a discussion on these limitations would be valuable. Otherwise, a comparison of GPU-/FPGA-accelerated TabuCol vs their FPGA-based vectorized approach would be a fairer benchmark.

Author Response

The reviewer brings up a fair point that the proposed 10,000x acceleration in FPGA based vectorized mapping compared to TabuCol heuristics on CPU might have come from choice of platform. However, tabu-search heuristics, commonly used in solving complex optimization problems, rely on sequential decision-making processes [R18-R19]. This sequential nature can limit the potential for parallel acceleration, as many heuristics are inherently designed to explore the solution space in a step-by-step manner. TabuCol heuristics used in this work is also sequential because it iteratively moves from one potential solution to another, using a tabu list to prevent revisiting recently explored solutions. This process maintains a memory of past solutions to guide future moves, influenced by the sequence of random decisions from previous iterations. As a result, it is challenging to parallelize without losing the benefits of its sequential exploration strategy.

As correctly pointed by the reviewer, there have been efforts to parallelize some steps of these algorithms to leverage the power of modern computing architectures like GPUs [R19-R21]. While GPU/FPGA implementation of the heuristic achieves modest speedups of up to an order of magnitude, these improvements are observed primarily for some problem instances. It holds true for our GPU implementation of TabuCol heuristics where we also achieve ~10X improvement for large problem instances and thus improving the scaling curve as shown in Figure R1.

Fig. R1 Time-to-solution (ns) comparison for TabuCol heuristics implemented on GPU and CPU.

For all the benchmarks, we updated the results with the GPU-based TabuCol heuristics implementation including Fig. 4 on page 17 and Table 1 on page 18.

Comment 3: The paper is very light on details about the truth table operator F. I suggest the authors to add more details on the F operator’s construction and size, either in the main text or supplementary materials.

Author Response

We thank the reviewer for the suggestion.

Algorithm-1 has been added for describing the construction of F operator on page 20. Further, additional description has been added page 4 paragraph 2.

Comment 4: The paper does not explicitly clarify whether the F operator must be redefined for each specific graph coloring problem or if it remains the same across different problem instances. If the operator is problem-dependent, it would be useful to discuss how its complexity scales and whether this introduces additional computational overhead compared to conventional mappings, which are typically O(Nq).

Author Response

We thank the reviewer for pointing this out. F operator needs to be defined for each problem described in Algorithm 1. The operator construction predominantly depends on the coloring constraint of the graph. This work uses single flip-Gibbs sampling to update the node, therefore, $N[\log_2(Q)]$ nodes (for N nodes Q color problem) need to be updated for vectorized mapping system for one complete iteration or time step. Thus, the time complexity for the proposed framework will scale with $O(N[\log_2(Q)])$. The time update for a single node update affects the prefactor of the time mentioned time complexity. Specifically, in probabilistic Ising/vectorized, this prefactor is contribution of multiplexer (states-weight multiplication), accumulator, sigmoid activation calculation and comparator delays. Owing

to binary nature of nodes, the accumulator dominates the node update timing [R26]. As a result, introducing higher-order multiplexers has minimal impact on the overall time complexity scaling.

The following lines are added on page 6 paragraph 2 line 6-8. “This process includes computing the F-operator, defined in Algorithm 1 which is implemented in hardware using truth tables or higher-order multiplexers, as illustrated in Fig. 2 b. The structure of this operator is largely determined by the graph’s coloring constraints.”

The following text has been added as paragraph 2 on page 7.

The FPGA architecture also uses single flip-Gibbs sampling to update the node, therefore, $N[\log_2(Q)]$ nodes need to be updated for vectorized mapping system for one complete iteration or time step. Thus, the time complexity for the proposed framework will scale with $O(N[\log_2(Q)])$. The time update for a single node update affects the prefactor of the time mentioned time complexity. Specifically, in probabilistic Ising/vectorized, this prefactor is contribution of multiplexer (states-weight multiplication), accumulator, sigmoid activation calculation and comparator delays. Owing to binary nature of nodes, the accumulator dominates the node update timing. As a result, introducing higher-order multiplexers has minimal impact on the overall time complexity scaling.

Minor comments:

Comment M1: To avoid potential misinterpretation, the authors may consider rewording 'brute force searching does not work' to 'brute force searching is impractical for large problem instances' to clarify that brute force is infeasible rather than fundamentally impossible for this class of problems.

Author Response

Thank you for the suggestion.

We have changed the line to: “The solution space grows exponentially with problem size, therefore making brute force searching impractical for large problem instances.” on page 1 paragraph 2.

Comment M2: I encourage the authors to specify whether LUT-based operations introduce any heuristic elements and how large the LUT should be for the 1024-spin IM.

Author Response

Thank you for pointing this out. LUT is used only to implement sigmoidal activation function $1/(1+e^{-x})$. LUT takes 8-bit inputs produced after neuron states-weight multiplication and accumulation (addition) and produces a 16-bit output. Therefore, this operation requires LUT with 2^8 (256) entries.

We have added the following lines on page 6 paragraph 2 line 8-12 for better clarity:

The accumulated product, represented with 8-bit precision, is passed through a sigmoid activation function that is implemented using a look-up table (LUT) containing 2^8 (256) entries. The LUT generates a 16-bit output, which is then compared with a 16-bit random number generated using a Linear Feedback Shift Register (LFSR) to get the updated node value.

Comment M3: To avoid potential confusion, I suggest specifying 'multi-state spin' instead of just 'multi-state' in the title and introduction since 'multi-state' can be perceived as a reference to different configurations of logical spins representing different solutions.

Author Response

We have changed the title to “Efficient Optimization Accelerator Framework for Multi-state Spin Ising Problems”.

Comment M4: The phrase "brute-force Ising approach" may cause some confusion, as it could be misinterpreted to mean that the Ising Machine itself operates via brute-force search. The authors could consider soften it with "constraint-heavy conventional Ising mapping" or clarify that this refers to the inefficiency introduced by excessive constraints in the mapping process, rather than the optimization method itself.

Author Response

Thank you for pointing this out.

We have changed “brute-force Ising approach” to “constraint-heavy conventional Ising mapping” for better clarity on page 7 paragraph 4.

References:

- [R1] R. M. R. Lewis, *A Guide to Graph Colouring - Algorithms and Applications* (Springer, Heidelberg, 2016).
- [R2] Schuetz, M. J. A., Brubaker, J. K., Zhu, Z. & Katzgraber, H. G. Graph coloring with physics inspired graph neural networks. *Phys. Rev. Res.* 4, 043131 (2022).
- [R3] Li, W. et al. Rethinking graph neural networks for the graph coloring problem. *ArXiv* (2022). 2208.06975.
- [R4] Hayato Goto et al., Combinatorial optimization by simulating adiabatic bifurcations in nonlinear Hamiltonian systems. *Sci. Adv.* 5, eaav2372(2019). DOI:10.1126/sciadv.aav2372.
- [R5] R. Ageron, T. Bouquet, and L. Pugliese, Simulated Bifurcation (SB) algorithm for Python, version 1.2.1, Nov. 2023. [Online]. Available: <https://github.com/bqth29/simulated-bifurcation-algorithm>.
- [R6] Lo, H., Moy, W., Yu, H. et al. An Ising solver chip based on coupled ring oscillators with a 48-node all-to-all connected array architecture. *Nat Electron* 6, 771–778 (2023). <https://doi.org/10.1038/s41928-023-01021-y>.
- [R7] Ryan Hamerly et al., Experimental investigation of performance differences between coherent Ising machines and a quantum annealer. *Sci. Adv.* 5, eaau0823 (2019). DOI:10.1126/sciadv.aau0823.
- [R8] M. -C. Li et al., "12.2 p-Circuits: Neither Digital Nor Analog," 2025 IEEE International Solid-State Circuits Conference (ISSCC), San Francisco, CA, USA, 2025, pp. 1-3, doi: 10.1109/ISSCC49661.2025.10904553.
- [R9] Patel, S., Canoza, P., Datar, A., Lu, S., Garg, C., & Salahuddin, S. (2024). PASS: An Asynchronous Probabilistic Processor for Next Generation Intelligence. *ArXiv*. <https://arxiv.org/abs/2409.10325>.
- [R10] A. K. McCallum, K. Nigam, J. Rennie, and K. Seymore, Automating the construction of internet portals with machine learning, *Information Retrieval* 3, 127 (2000).
- [R11] P. Sen, G. Namata, M. Bilgic, L. Getoor, B. Galligher, and T. Eliassi-Rad, Collective classification in network data, *AI magazine* 29, 93 (2008).
- [R12] Namata, G., London, B., Getoor, L. & Huang, B. Query-driven active surveying for collective classification. In *Proceedings of the 10th International Workshop on Mining and Learning with Graphs*, vol. 8, 249–256 (2012).
- [R13] Pierre Brémaud. "Gibbs Fields and Monte Carlo Simulation". In: *Markov Chains*. New York, NY: Springer New York, 1999, pp. 253–322. doi: 10.1007/978-1-4757-3124-8_{ }7. url: http://link.springer.com/10.1007/978-1-4757-3124-8_7.
- [R14] Ackley, D. H., Hinton, G. E., & Sejnowski, T. J. (1985). A learning algorithm for Boltzmann machines. *Cognitive Science*, 9(1), 147–169.
- [R15] K. Y. Camsari, R. Faria, B. M. Sutton, and S. Datta, "Stochastic p-Bits for Invertible Logic," *Phys. Rev. X* 7, 031014 (2017). DOI: 10.1103/PhysRevX.7.031014.
- [R16] Borders, W.A., Pervaiz, A.Z., Fukami, S. et al. Integer factorization using stochastic magnetic tunnel junctions. *Nature* 573, 390–393 (2019). <https://doi.org/10.1038/s41586-019-1557-9>.
- [R17] Si, J., Yang, S., Cen, Y. et al. Energy-efficient superparamagnetic Ising machine and its application to traveling salesman problems. *Nat Commun* 15, 3457 (2024). <https://doi.org/10.1038/s41467-024-47818-z>.
- [R18] Hertz, A. & de Werra, D. Using tabu search techniques for graph coloring. *Computing* 39, 345–351 (1987).
- [R19] Zhu, W., Curry, J., & Marquez, A. (2009). SIMD tabu search for the quadratic assignment problem with graphics hardware acceleration. *International Journal of Production Research*, 48(4), 1035–1047. <https://doi.org/10.1080/00207540802555744>.
- [R20] Clara Novoa, Apan Qasem, and Abhilash Chaparala. 2015. A SIMD tabu search implementation for solving the quadratic assignment problem with GPU acceleration. In *Proceedings of the 2015 XSEDE Conference: Scientific Advancements Enabled by Enhanced Cyberinfrastructure (XSEDE '15)*. Association for Computing Machinery, New York, NY, USA, Article 13, 1–8. <https://doi.org/10.1145/2792745.2792758>.
- [R21] K. Kanazawa, "Accelerating Swap-Based Tabu Search for Solving Maximum Clique Problems on FPGA," 2019 IEEE Intl Conf on Parallel & Distributed Processing with Applications, Big Data & Cloud Computing, Sustainable Computing & Communications, Social Computing & Networking (ISPA/BDCLOUD/SocialCom/SustainCom), Xiamen, China, 2019, pp. 1033-1040, doi: 10.1109/ISPA-BDCLOUD-SustainCom-SocialCom48970.2019.00148.
- [R22] Trick, M. COLOR Dataset (2002). Accessed: 2024-09-29.
- [R23] K. Yamamoto et al., "7.3 STATICA: A 512-Spin 0.25M-Weight Full-Digital Annealing Processor with a Near-Memory All-Spin-Updates-at-Once Architecture for Combinatorial Optimization with Complete Spin-Spin Interactions," 2020 IEEE International Solid-State Circuits Conference - (ISSCC), San Francisco, CA, USA, 2020, pp. 138-140, doi: 10.1109/ISSCC19947.2020.9062965.
- [R24] Y. Su, H. Kim and B. Kim, "31.2 CIM-Spin: A 0.5-to-1.2V Scalable Annealing Processor Using Digital Compute-In-Memory Spin Operators and Register-Based Spins for Combinatorial Optimization Problems," 2020 IEEE International Solid-State Circuits Conference - (ISSCC), San Francisco, CA, USA, 2020, pp. 480-482, doi: 10.1109/ISSCC19947.2020.9062938.

- [R25] Y. Zhou, G. Su, J. Zhou, L. Liao and Z. Chen, "A Compute-in-Memory Annealing Processor With Interaction Coefficient Reuse and Sparse Energy Computation for Solving Combinatorial Optimization Problems," in *IEEE Journal of Solid-State Circuits*, vol. 59, no. 9, pp. 3094-3105, Sept. 2024, doi: 10.1109/JSSC.2024.3376410.
- [R26] Patel, S., Canoza, P. & Salahuddin, S. Logically synthesized and hardware-accelerated restricted Boltzmann machines for combinatorial optimization and integer factorization. *Nat Electron* 5, 92–101 (2022). <https://doi.org/10.1038/s41928-022-00714-0>

Re: NCOMMS-24-85810A

Efficient Optimization Accelerator Framework for Multistate Spin Ising Problems

Chirag Garg, Sayeef Salahuddin

General Comment:

Both reviewer 1 and reviewer 2 have asked a question about CPU/GPU/FPGA implementation of Heuristics vs the proposed method. We provided a point-by-point answer to their questions. But given the commonality of this particular question, we added a paragraph in page 7. The paragraph is given below:

One question that may arise is: is the acceleration achieved for the proposed method over Heuristics solely due to the fact that the vectorized method was implemented on FPGA? In this regard, we note here that the Heuristics take advantage of sequential strategies that do not scale well on parallel architecture. This has been studied extensively in the literature. For example, in Extended Data Fig. 10(a) we show a comparison between CPU and GPU implementations of Tabucol. It is clearly seen that when the problem size increases, the GPU implementation provides virtually no acceleration. By contrast GPU implementation of the proposed vectorized method shows large acceleration as shown in Extended Data Fig. 10(b). This underscores a strength of the proposed method that makes it amenable for scaling on specialized hardware such as the FPGA.

We also note that the fact that Heuristics do not scale well on parallelized systems such as GPU/FPGA is quite well known and has been extensively discussed in literature [R5-R7].

For the reviewers' convenience Fig. SI 10 is pasted below:

Figure S10: Performance comparison between CPU and GPU implementation of Tabucol heuristics and Vectorized framework.

Reviewer – 1

The authors have made revisions based on the reviewers' comments; however, several issues still need to be addressed: We thank the reviewer for providing the review of the paper. Below, we provide a detailed response to reviewer's comment.

Comment 1: The authors have added comparisons with Tabu-search heuristics, machine learning approaches, and a physics-based solver. However, these comparisons are not conducted on a fair basis. For instance, the Tabucol algorithm is implemented on a GPU, the machine learning-based results are taken from the literature, and the vectorized method is implemented on an FPGA. To ensure a fair evaluation, all methods should be tested under the same hardware platform or comparable conditions.

Author Response

Note that Table 1 and Fig. 4c in our previous round reported results from all methods (Tabucol, machine learning and vectorized) on the **same hardware platform**, which is the GPU. Additionally, we make the point that the vectorized method scales very well on FPGA, providing a further acceleration. Note that, it is not given that every algorithm will scale well on every platform. The fact that the proposed method can be accelerated on FPGA is a positive attribute of the vectorized method.

In the previous round, we further explained, in answering Reviewer 4's question on the same topic, that Heuristic algorithms are not typically tested on specialized platform such as GPU or FPGA. For Reviewer 1's convenience, we present the discussion here again:

Heuristic algorithms rely on sequential search strategies. As a result, they do not scale very well on parallel hardware like GPUs or FPGAs [R5–R7]. Consequently, several recent studies [R3–R4] have benchmarked their optimization solvers against CPU-based heuristics, despite the known architectural mismatch. In other words, the way we presented our results is the norm, and not an exception.

Comment 2: The area comparison cannot be solely based on the number of physical nodes, as the hardware complexity per node varies significantly across different implementations. The authors should provide synthesis area metrics (e.g., gate count or layout area) for each method to allow a more meaningful comparison.

Author Response

Owing to binary nature of nodes in large scale, densely connected Ising machines, the area is predominantly dominated by accumulator [R8]. Since the accumulator's complexity scales proportionally with the number of physical nodes, we believe that using physical node count provides a reasonable and meaningful basis for area comparison.

Additionally, for a more rigorous comparison, we have synthesized both the probabilistic Ising and vectorized architectures using Cadence Genus synthesis tool and TSMC 28 PDK standard cells. Both designs are synthesized to support a 256-node, 16-color graph coloring problem under a 90 MHz clock constraint that matches our FPGA implementation setup. The post-synthesis results below clearly shows that the probabilistic Ising architecture requires significantly more area compared to vectorized implementation, thereby supporting our area comparison.

Design	Cell Count	Cell Area (μm^2)
Ising	127094	124348.014
Vectorized	21188	20511.288

Comment 3: The authors have included results for solving the Traveling Salesman Problem (TSP). However, addressing instances with only 4 to 14 cities is relatively trivial. The authors are encouraged to tackle more challenging TSP instances and provide detailed descriptions of the simulation setup, including the choice of hyperparameters and any optimization strategies used.

Author Response

We appreciate the reviewer's suggestion regarding the inclusion of more challenging TSP instances. We want to emphasize the fact that we only brought in the TSP results in response to to Reviewer-1's Comment 9 in the previous

round to highlight the generality of our proposed mapping approach across different classes of combinatorial problems. But this paper is not about TSP problem itself.

Graph coloring problem, given its wide applicability in areas such as VLSI layout decomposition, resource allocation, logic minimization, and industrial scheduling, is considered an important class of problem in its own right, and many previous works [such R9–R11] have been solely dedicated on this problem itself.

Similarly, TSP problem is an important class of problems and a proper treatment of this problem, including solving large instances, will have to be done through dedicated studies solely on this problem as we find in literature [R12-R14]. Our results on TSP simply shows that the same vectorized mapping technique works for a different class of problem, establishing the generality of the method.

Comment 4: The core contribution of this work lies in the proposed encoding method for multi-state Ising problems. However, the significance of this contribution appears limited and should be more critically evaluated or better substantiated. Overall, the authors are expected to provide more comprehensive details on the simulation procedures to ensure reproducibility. In addition, they should be more cautious with strong claims such as “surpassing the capabilities of any existing Ising method,” unless they are rigorously supported by fair and extensive comparisons.

Author Response

While the proposed encoding method is a core contribution, its significance lies in enabling efficient hardware acceleration, particularly on FPGAs, where binary node operations can be effectively leveraged. Unlike many heuristics, our approach is amenable to such hardware acceleration, which broadens its applicability.

Regarding evaluation, we believe the method has been assessed fairly on graph coloring problem including real-world datasets such as citation graphs and demonstrated its applicability on other complex problem like TSP. The comprehensive details on simulation procedures are present in supplementary section 1-4, Algorithm 1-3 and supplementary figure S1-S8.

Additionally, we would like to mention, based on the review, we already removed the phrase “surpassing the capabilities of any existing Ising method”. This phrase is not present in the revised manuscript submitted in the last round.

Reviewer – 2

We would like to begin by thanking the authors for their thoughtful response to the initial round of review. The manuscript has improved in several respects, and we appreciate the effort that has gone into the revision. Nevertheless, we remain concerned about the strength and accuracy of certain claims made throughout the paper. In several key areas, the assertions appear overstated and may mislead readers about the true performance and novelty of the proposed approach.

We thank the reviewer for providing the review of the paper. Below, we provide a detailed response to reviewer’s comment.

In the abstract, the authors state that their method achieves accuracy comparable to state-of-the-art heuristics and machine learning algorithms, with significant improvements over Ising-based methods. While this is partially supported by Tables 1 and 2, the claim fails to acknowledge a critical trade-off. As illustrated in Figures 4c and 4d, the vectorized method exhibits comparable time-to-solution (TTS) and energy-to-solution to approaches such as Simulated Bifurcation and TabuCol. This suggests that the improved accuracy comes at the cost of longer runtimes, which offsets the overall performance gains. We strongly encourage the authors to revise this claim to more accurately reflect the trade-offs involved and to explicitly discuss this point when introducing Figure 4c.

Author Response

The authors state that their method achieves accuracy comparable to state-of-the-art heuristics and machine learning algorithms, with significant improvements over Ising-based methods. As illustrated in Figures 4c and 4d, the vectorized method exhibits comparable time-to-solution (TTS) and energy-to-solution to approaches such as Simulated Bifurcation and Tabucol. This suggests that the improved accuracy comes at the cost of longer runtimes, which offsets the overall performance gains.

Actually, from Fig 4c and 4d, one sees that the proposed method achieves much more improved accuracy than probabilistic Ising and Simulated Bifurcation methods, while needing comparable TTS and Energy-to-solution.

If the referee's comment is solely about Tabucol Heuristics, we refer the referee to our answer to their next comment below.

It is also important to clarify that the reported performance gains are only observed in the FPGA implementation. Figures 4c and 4d indicate that the performance advantage is largely due to the hardware acceleration provided by the FPGA platform, rather than the proposed encoding scheme itself. Consequently, the claim of a “10,000× performance acceleration over heuristics” is misleading in its current form and requires substantial revision.

Although the reviewer does not explicitly mention it, it seems to us that there is an implicit assumption that Tabucol will accelerate similarly to the other algorithms if implemented on GPU or FPGA. **This is not correct.** Heuristics mostly rely on sequential searches that do not scale on parallel architecture. This is in fact quite well known in the community (for example see [R5-R7] where these considerations have been discussed). Because of this reason, it is customary in literature to show performance of Heuristics only on CPU.

Nonetheless because the reviewer has asked this question, we have now included a Supplementary figure S10 (reproduced below for convenience.) where we compare Tabucol TTS for CPU and GPU (S10(a)). As it can be seen, once the problem size increases there is minimal improvement in TTS by going from CPU to GPU. This is simply due to the reason described above. Given that there is no acceleration over GPU, there is no real reason to then implement it in FPGA.

By contrast, the for the proposed method, GPU acceleration shows 10X-100X improvement over CPU (Fig. S10 (b)), confirming that the proposed method can take advantage of the parallel architecture. This motivates the implementation on FPGA.

Therefore, the fact that we have compared the FPGA implementation of the proposed vectorized method with Heuristics run on GPU is not an unfair comparison – rather this is the norm [R3–R4]. Additionally, it emphasizes the strength of the proposed method in terms of its amenability for parallelization. We have now included a new paragraph in page 7 to highlight it.

Figure S10: Performance comparison between CPU and GPU implementation of Tabucol heuristics and Vectorized framework.

Similarly, the statement that parallel tempering reduces coloring error by 50% should be contextualized. This effect is not unique to the proposed method—it applies equally to the Ising-based techniques. Presenting it as a specific advantage of the proposed framework is therefore inaccurate and should be reframed accordingly.

Author Response

Thank you for pointing this out.

We have changed the modified the statement in abstract to:

Additionally, we adopt parallel tempering with the proposed framework to further reduce the coloring error by up to 50% compared to the Gibbs sampling algorithm.

As stated in our previous review report, in our view, the paper's most promising contribution lies in the novel encoding strategy, which may be of real interest specifically to the Ising machine and optimization community. However, the use of FPGAs for acceleration is well established, and Figures 4c and 4d suggest that the proposed method's performance and energy efficiency are comparable to existing techniques. While the encoding approach holds potential, the overall results do not meet the threshold of novelty or broad impact expected for publication in Nature Communications. A more specialized journal may be more appropriate if the claims are revised to more accurately reflect the data.

Author Response

As we mentioned above, we do not understand the comment: "Figures 4c and 4d suggest that the proposed method's performance and energy efficiency are comparable to existing techniques"

Fig 4c and 4d clearly shows that the proposed methods performance and energy efficiency is much better.

However, the use of FPGAs for acceleration is well established,

Again, this is not correct. Not every algorithm accelerates on FPGA. We speculate (again the reviewers have not explicitly mentioned it) that the reviewer believes that Heuristics would have also accelerated on FPGA. If indeed this is the point that the reviewer is making, then it is incorrect and the fact that Heuristics don't accelerate on parallelized systems like GPU and FPGA is very well known (as we discussed above).

As stated in our previous review report, in our view, the paper's most promising contribution lies in the novel encoding strategy, which may be of real interest specifically to the Ising machine and optimization community.

.....

While the encoding approach holds potential, the overall results do not meet the threshold of novelty or broad impact expected for publication in Nature Communications. A more specialized journal may be more appropriate if the claims are revised to more accurately reflect the data.

This is a very surprising conclusion. The reviewers say that the results may of 'real interest to Ising machine and optimization community.' Optimization permeates every important field of science. But an important result in optimization does not have wide interest? What about the large number of papers that Nature family journals have published on both Ising machine and Optimization in the recent years?

Overall, we feel that the reviewer actually provides the perfect rationale for publication in Nature Communications. They mentioned that the results are of 'real interest' to the Ising machine and optimization community – this should suffice for the rationale of publication.

We provide the following detailed suggestions aimed at improving clarity, accuracy, and reproducibility:

Comment 1: Equation (2): The factor B appears to be missing, and both A and B are only defined in Supplementary Note 4. These definitions should be included in the main text.

Author Response

Thank you for pointing this out. We have modified the Eq. 2 on page 3 and defined both A & B parameters on line 2 paragraph 3 page 3 in the main text.

Comment 2: Hyperparameter optimization: The search space is currently limited to a 2D scan over A and B, omitting key parameters such as temperature. This may result in suboptimal performance for the Ising machine and an unfair comparison with the proposed method. Since only the ratio B/A matters (due to energy rescaling), a 1D scan would

suffice. The computational resources saved could then be redirected to a broader search that includes temperature and other relevant parameters.

Author Response

We thank the reviewer for raising this point. While it is true that temperature (T) is a relevant parameter in many Ising models, our update rule (Eq. R1 equivalent of Eq. 4) depends only on the ratio H / T , as shown in Eq. R2. Since both terms in the energy function scale with $1/T$, fixing T while scanning over A and B is equivalent to scanning over the normalized quantities A/T and B/T. Thus, our 2D grid search over A and B with fixed T effectively explores the same hyperparameter landscape as a search over B/A and T at fixed A. Therefore, our hyperparameter optimization captures the relevant dynamics, and the presented results reflect optimal settings within this equivalence.

$$P(s_{ik} = 1) = \sigma \left(-\frac{d(H/T)}{ds_{ik}} \right) \dots\dots\dots (R1)$$

$$\frac{H}{T} = \frac{A}{T} \sum_{i,j \in E} \sum_{k=1}^q s_{ik} s_{jk} + \frac{B}{T} \sum_i \left(1 - \sum_{k=1}^q s_{ik} \right)^2 \dots\dots\dots (R2)$$

Comment 3: Definition of E: Just below Equation (2), E is incorrectly defined as the edge between nodes i and j. In fact, E denotes the set of all edges in the graph and should be corrected accordingly.

Author Response

Thank you for bringing this up. We have modified the line 1-2 paragraph 3 page 3 reflecting this change.

Comment 4: Simulated Bifurcation: The description on page 5 lacks sufficient detail. Important parameters are not specified, which impedes reproducibility. Given the sensitivity of this method to parameter choices, additional detail on the optimization process is necessary. Furthermore, its results should be included in Table 2 for completeness.

Author Response

Thank you for the suggestion. We have added details about hyperparameter optimization for simulated bifurcation method on lines 13-17 paragraph 3 page 10 and supplementary Figure S2 on page 28. Additionally, the results of simulated bifurcation have been added in Table 2 on page 19.

Comment 5: Notation consistency: In Supplementary Note 1, the inverse temperature $\beta \sim 1/T$ is used without definition, whereas Supplementary Note 2 refers directly to T. Consistent notation throughout the supplementary materials is essential to avoid confusion.

Author Response

Thank you for the suggestion. To improve the clarity, we have revised the relevant equation using β in supplementary section 1 on page 9.

Reviewer – 4

The authors have improved the manuscript significantly, adding additional benchmarks and expanding the number of instances used for existing benchmarks. The reviewers' concerns and comments were properly addressed.

A new section devoted to benchmarking with the citation graph dataset has been added in the Supplementary Materials. The authors have also added additional benchmarks in Table-1 on page 18 to confirm that the solution accuracy improves mainly due to the vectorized approach.

A grid search parameter optimization has been introduced as suggested. The authors have also provided necessary details about the construction of the truth table operator F in the main text and added an Algorithm-1.

I now fully support the publication of authors' manuscript in Nature Communications.

Author Response

We sincerely appreciate the reviewer's thoughtful comments and are pleased to hear that the revisions have addressed their concerns satisfactorily. We are grateful for the reviewer's support of our manuscript and recognition of the improvements made, including the expanded benchmarks, and the clarification of key methodological detail.

References

- [R1] Schuetz, M. J. A., Brubaker, J. K., Zhu, Z. & Katzgraber, H. G. Graph coloring with physics inspired graph neural networks. *Phys. Rev. Res.* 4, 043131 (2022).
- [R2] Hertz, A. & de Werra, D. Using tabu search techniques for graph coloring. *Computing* 39, 345–351 (1987).
- [R3] Cilasun, H., Moy, W., Zeng, Z. et al. A coupled-oscillator-based Ising chip for combinatorial optimization. *Nat Electron* 8, 537–546 (2025). <https://doi.org/10.1038/s41928-025-01393-3>.
- [R4] Pedretti, G., Böhm, F., Bhattacharya, T. et al. Solving Boolean satisfiability problems with resistive content addressable memories. *npj Unconv. Comput.* 2, 7 (2025). <https://doi.org/10.1038/s44335-025-00020-w>.
- [R5] Zhu, W., Curry, J., & Marquez, A. (2009). SIMD tabu search for the quadratic assignment problem with graphics hardware acceleration. *International Journal of Production Research*, 48(4), 1035–1047. <https://doi.org/10.1080/00207540802555744>.
- [R6] Clara Novoa, Apan Qasem, and Abhilash Chaparala. 2015. A SIMD tabu search implementation for solving the quadratic assignment problem with GPU acceleration. In *Proceedings of the 2015 XSEDE Conference: Scientific Advancements Enabled by Enhanced Cyberinfrastructure (XSEDE '15)*. Association for Computing Machinery, New York, NY, USA, Article 13, 1–8. <https://doi.org/10.1145/2792745.2792758>.
- [R7] K. Kanazawa, "Accelerating Swap-Based Tabu Search for Solving Maximum Clique Problems on FPGA," 2019 IEEE Intl Conf on Parallel & Distributed Processing with Applications, Big Data & Cloud Computing, Sustainable Computing & Communications, Social Computing & Networking (ISPA/BDCLOUD/SocialCom/SustainCom), Xiamen, China, 2019, pp. 1033-1040, doi: 10.1109/ISPA-BDCLOUD-SustainCom-SocialCom48970.2019.00148.
- [R8] Patel, S., Canoza, P. & Salahuddin, S. Logically synthesized and hardware-accelerated restricted Boltzmann machines for combinatorial optimization and integer factorization. *Nat Electron* 5, 92–101 (2022). <https://doi.org/10.1038/s41928-022-00714-0>.
- [R9] Yue, W., Zhang, T., Jing, Z. et al. A scalable universal ising machine based on interaction-centric storage and compute-in-memory. *Nature Electronics* (2024).
- [R10] Inaba, K., Inagaki, T., Igarashi, K. et al. Potts model solver based on hybrid physical and digital architecture. *Communications Physics* 5, 137 (2022).
- [R11] Whitehead, W., Nelson, Z., Camsari, K. et al. Cmos-compatible ising and potts annealing using single-photon avalanche diodes. *Nature Electronics* 6, 1009–1019 (2023).
- [R12] Si, J., Yang, S., Cen, Y. et al. Energy-efficient superparamagnetic Ising machine and its application to traveling salesman problems. *Nat Commun* 15, 3457 (2024). <https://doi.org/10.1038/s41467-024-47818-z>.
- [R13] A. Dan, R. Shimizu, T. Nishikawa, S. Bian and T. Sato, "Clustering Approach for Solving Traveling Salesman Problems via Ising Model Based Solver," 2020 57th ACM/IEEE Design Automation Conference (DAC), San Francisco, CA, USA, 2020, pp. 1-6, doi: 10.1109/DAC18072.2020.9218695.
- [R14] Qichao Tao and Jie Han. 2022. Solving traveling salesman problems via a parallel fully connected ising machine. In *Proceedings of the 59th ACM/IEEE Design Automation Conference (DAC '22)*. Association for Computing Machinery, New York, NY, USA, 1123–1128. <https://doi.org/10.1145/3489517.3530595>.

Re: NCOMMS-24-85810B
Efficient Optimization Accelerator Framework for Multistate Spin Ising Problems
Chirag Garg, Sayeef Salahuddin

Reviewer – 1

The authors have addressed my questions in the revised manuscript. I have no further comments.

We sincerely thank the reviewer for the thoughtful comments and are glad that our revisions have satisfactorily addressed the comments.

Reviewer – 2

We appreciate the authors' efforts in addressing our previous comments. It has now become clear to us that there is a misunderstanding surrounding the term heuristics. We interpret heuristics broader as compared to the authors and also include probabilistic Ising machines and simulated bifurcation methods under this category.

Therefore, we had an issue with the “10,000× performance acceleration over heuristics” in the abstract, as the comparison is made solely against TabuCol, and not against the other methods such as simulated bifurcation or probabilistic Ising machines. If our interpretation of what constitutes a heuristic differs from that of the authors, it is likely that other readers may also interpret this differently. Therefore, we think it appropriate if the abstract would be rewritten to be more precise and to be clear that it is a 10,000 acceleration when comparing TabuCol on GPU and the FPGA implementation of the authors.

Author Response

We sincerely appreciate the reviewer’s acknowledgment of our efforts in addressing the previous comments. We also thank the reviewer for clarifying the broader interpretation of heuristics, including probabilistic Ising machines and simulated bifurcation methods.

We have rewritten the statement in abstract to be more precise:

We also design a 1024-neuron all-to-all connected probabilistic Ising accelerator on FPGA with the proposed approach that shows ~10000× performance acceleration compared to GPU-based TabuCol heuristics.

Similarly, the statement in the Discussion Section has been updated on page 8:

Overall, the hardware implementation consumes 5W power and achieves approximately ~10000× speedup compared to TabuCol heuristics.

Additionally, we agree with the authors that it is true that not necessarily all methods will benefit from implementation on the FPGA. Nevertheless, we would like to refer the authors to [R1], which claims that simulated bifurcation can also benefit from an implementation on an FPGA rather than a GPU. Of course, the numbers in this paper cannot necessarily be directly compared with the current manuscript. We do believe that other readers will have the same reaction as us that Figure 4 warrants an apples to apples comparison of all the different methods. Therefore, we think it worthwhile that at least a discussion is included that the performance gap between the proposed vectorized approach on FPGA and e.g. simulated bifurcation on FPGA can be narrower than the factor 10,000.

Author Response

We thank the reviewer for highlighting the potential of FPGA acceleration for simulated bifurcation methods [R1]. We have updated the manuscript to clarify that Ising mapping-based solvers, including probabilistic Ising and simulated bifurcation methods, can benefit from FPGA acceleration.

We have added the following lines on page 7, paragraph 3, line 6-10:

“Similarly, Ising mapping-based solvers, including Probabilistic Ising and Simulated Bifurcation, can benefit from FPGA acceleration [R1], potentially narrowing the performance gap with our FPGA implementation of the vectorized

mapping. Nevertheless, for problems with more than 100 nodes, these solvers may continue to yield sub-optimal solutions.”

References

[R1] K. Tatsumura, A. R. Dixon and H. Goto, "FPGA-Based Simulated Bifurcation Machine," 2019 29th International Conference on Field Programmable Logic and Applications (FPL), Barcelona, Spain, 2019, pp. 59-66, doi: 10.1109/FPL.2019.00019.